# Thresholding Bandit with Optimal Aggregate Regret

**Chao Tao**
Computer Science Department
Indiana University at Bloomington

**Saúl A. Blanco**
Computer Science Department
Indiana University at Bloomington

**Jian Peng**
Computer Science Department
University of Illinois at Urbana-Champaign

**Yuan Zhou**
Computer Science Department, Indiana University at Bloomington
Department of ISE, University of Illinois at Urbana-Champaign

## Abstract

We consider the thresholding bandit problem, whose goal is to find arms of mean rewards above a given threshold $\theta$, with a fixed budget of $T$ trials. We introduce LSA, a new, simple and anytime algorithm that aims to minimize the aggregate regret (or the expected number of mis-classified arms). We prove that our algorithm is instance-wise asymptotically optimal. We also provide comprehensive empirical results to demonstrate the algorithm's superior performance over existing algorithms under a variety of different scenarios.

## 1  Introduction

The stochastic *Multi-Armed Bandit* (MAB) problem has been extensively studied in the past decade (Auer, 2002; Audibert et al., 2010; Bubeck et al., 2009; Gabillon et al., 2012; Karnin et al., 2013; Jamieson et al., 2014; Garivier and Kaufmann, 2016; Chen et al., 2017). In the classical framework, at each trial of the game, a learner faces a set of $K$ *arms*, pulls an arm and receives an unknown stochastic reward. Of particular interest is the *fixed budget* setting, in which the learner is only given a limited number of total pulls. Based on the received rewards, the learner will recommend the best arm, i.e., the arm with the highest mean reward. In this paper, we study a variant of the MAB problem, called the *Thresholding Bandit Problem* (TBP). In TBP, instead of finding the best arm, we expect the learner to identify all the arms whose mean rewards $\theta_i$ ($i \in \{1, 2, 3, \ldots, K\}$) are greater than or equal to a given threshold $\theta$. This is a very natural setting with direct real-world applications to active binary classification and anomaly detection (Locatelli et al., 2016; Steinwart et al., 2005).

In this paper, we propose to study TBP under the notion of *aggregate regret*, which is defined as the expected number of errors after $T$ samples of the bandit game. Specifically, for a given algorithm $\mathbb{A}$ and a TBP instance $I$ with $K$ arms, if we use $e_i$ to denote the probability that the algorithm makes an incorrect decision corresponding to arm $i$ after $T$ rounds of samples, the aggregate regret is defined as $\mathcal{R}^{\mathbb{A}}(I; T) \overset{\text{def}}{=} \sum_{i=1}^{K} e_i$. In contrast, most previous works on TBP aim to minimize the *simple regret*, which is the probability that at least one of the arms is incorrectly labeled. Note that the definition of aggregate regret directly reflects the overall classification accuracy of the TBP algorithm, which is more meaningful than the simple regret in many real-world applications. For example, in the crowdsourced binary labeling problem, the learner faces $K$ binary classification tasks, where each task $i$ is associated with a latent true label $z_i \in \{0, 1\}$, and a latent soft-label $\theta_i$. The soft-label $\theta_i$ may be used to model the *labeling difficulty/ambiguity* of the task, in the sense that $\theta_i$ fraction

of the crowd will label task $i$ as $1$ and the rest labels task $i$ as $0$. The crowd is also assumed to be *reliable*, i.e., $z_i = 1$ if and only if $\theta_i \geq \frac{1}{2}$. The goal of the crowdsourcing problem is to sequentially query a random worker from the large crowd about his/her label on task $i$ for a budget of $T$ times, and then label the tasks with as high (expected) accuracy as possible. If we treat each of the binary classification task as a Bernoulli arm with mean reward $\theta_i$, then this crowdsourced problem becomes aggregate regret minimization in TBP with $\theta = \frac{1}{2}$. If a few tasks are extremely ambiguous (i.e., $\theta_i \to \frac{1}{2}$), the simple regret would trivially approach $1$ (i.e., every algorithm would almost always fail to correctly label all tasks). In such cases, however, a good learner may turn to accurately label the less ambiguous tasks and still achieve a meaningful aggregate regret.

A new challenge arising for the TBP with aggregate regret is how to balance the exploration for each arm given a fixed budget. Different from the exploration vs. exploitation trade-off in the classical MAB problems, where exploration is only aimed for finding the best arm, the TBP expects to maximize the accuracy of the classification of *all* arms. Let $\Delta_i \overset{\text{def}}{=} |\theta_i - \theta|$ be the *hardness* parameter or *gap* for each arm $i$. An arm with smaller $\Delta_i$ would need more samples to achieve the same classification confidence. A TBP learner faces the following dilemma – whether to allocate samples to determine the classification of one hard arm, or use it for improving the accuracy of another easier arm.

**Related Work.**    Since we focus on the TBP problem in this paper, due to limit of the space, we are sorry for not being able to include the significant amount of references to other MAB variants.

In a previous work (Locatelli et al., 2016), the authors introduced the APT (Anytime Parameter-free Thresholding) algorithm with the goal of simple regret minimization. In this algorithm, a precision parameter $\epsilon$ is used to determine the tolerance of errors (a.k.a. the indifference zone); and the APT algorithm only attempts to correctly classify the arms with hardness gap $\Delta_i > \epsilon$. This variant goal of simple regret partly alleviates the trivialization problem mentioned previously because of the extremely hard arms. In details, at any time $t$, APT selects the arm that minimizes $\sqrt{T_i(t)}\widehat{\Delta}_i(t)$, where $T_i(t)$ is the number of times arm $i$ has been pulled until time $t$, $\widehat{\Delta}_i(t)$ is defined as $|\widehat{\theta}_i(t) - \theta| + \epsilon$, and $\widehat{\theta}_i(t)$ is the empirical mean reward of arm $i$ at time $t$. In their experiments, Locatelli et al. (2016) also adapted the UCBE algorithm from (Audibert et al., 2010) for the TBP problem and showed that APT performs better than UCBE.

When the goal is to minimize the aggregate regret, the APT algorithm also works better than UCBE. However, we notice that the choice of precision parameter $\epsilon$ has significant influence on the algorithm's performance. A large $\epsilon$ makes sure that, when the sample budget is limited, the APT algorithm is not intrigued by a hard arm to spend overwhelmingly many samples on it without achieving a confident label. However, when the sample budget is ample, a large $\epsilon$ would also prevent the algorithm from making enough samples for the arms with hardness gap $\Delta_i < \epsilon$. Theoretically, the optimal selection of this precision parameter $\epsilon$ may differ significantly across the instances, and also depends on the budget $T$. In this work, we propose an algorithm that does not require such a precision parameter and demonstrates improved robustness in practice. Furthermore, a simple corollary of our main theorem (Theorem 1) shows that, for the simple regret with no indifference zone ($\epsilon = 0$), our LSA algorithm achieves the optimality up to a $\ln K$ factor in the budget $T$ compared with APT(0). We attach experimental results in Appendix F.1 to show that LSA performs better than APT(0) towards the simple regret objective.

Another natural approach to TBP is the uniform sampling method, where the learner plays each arm the same number of times (about $T/K$ times). In Appendix C, we show that the uniform sampling approach may need $\Omega(K)$ times more budget than the optimal algorithm to achieve the same aggregate regret.

Finally, Chen et al. (2015) proposed the optimistic knowledge gradient heuristic algorithm for budget allocation in crowdsourcing binary classification with Beta priors, which is closely related to the TBP problem in the Bayesian setting.

**Our Results and Contributions.**    Let $\mathcal{R}^{\mathbb{A}}(I; T)$ denote the aggregate regret of an instance $I$ after $T$ time steps. Given a sequence of hardness parameters $\Delta_1, \Delta_2, \ldots, \Delta_K$, assume $\mathcal{I}_{\Delta_1, \ldots, \Delta_K}$ is the class of all $K$-arm instances where the gap between $\theta_i$ of the $i$-th arm and the threshold $\theta$ is $\Delta_i$, and let

$$\mathrm{OPT}(\{\Delta_i\}_{i=1}^K, T) \stackrel{\text{def}}{=} \inf_{\mathbb{A}} \sup_{I \in \mathcal{I}_{\Delta_1,\ldots,\Delta_K}} \mathcal{R}^{\mathbb{A}}(I;T) \tag{1}$$

be the minimum possible aggregate regret that any algorithm can achieve among all instances with the given set of gap parameters. We say an algorithm $\mathbb{A}$ is *instance-wise asymptotically optimal* if for every $T$, any set of gap parameters $\{\Delta_i\}_{i=1}^K$, and any instance $I \in \mathcal{I}_{\Delta_1,\ldots,\Delta_K}$, it holds that

$$\mathcal{R}^{\mathbb{A}}(I;T) \leq O(1) \cdot \mathrm{OPT}(\{\Delta_i\}_{i=1}^K, \Omega(T)). \tag{2}$$

While it may appear that a constant factor multiplied to $T$ can affect the regret if the optimal regret is an exponential function of $T$, we note that our definition aligns with major multi-armed bandit literature (e.g., fixed-budget best arm identification (Gabillon et al., 2012; Carpentier and Locatelli, 2016) and thresholding bandit with simple regret (Locatelli et al., 2016)). Indeed, according to our definition, if the universal optimal algorithm requires a budget of $T$ to achieve $\epsilon$ regret, an asymptotically optimal algorithm requires a budget of only $T$ multiplying some constant to achieve the same order of regret. On the other hand, if one wishes to pin down the optimal constant before $T$, even for the single arm case, it boils down to the question of the optimal (and distribution dependent) constant in the exponent of existing concentration bounds such as Chernoff Bound, Hoeffding's Inequality, and Bernstein Inequalities, which is beyond the scope of this paper.

We address the challenges mentioned previously and introduce a simple and elegant algorithm, the Logarithmic-Sample Algorithm (LSA). LSA has a very similar form as the APT algorithm in (Locatelli et al., 2016) but introduces an additive term that is proportional to the logarithm of the number of samples made to each arm in order to more carefully allocate the budget among the arms (see Line 4 of Algorithm 1). This logarithmic term arises from the optimal sample allocation scheme of an offline algorithm when the gap parameters are known beforehand. The log-sample additive term of LSA can be interpreted as an incentive to encourage the samples for arms with bigger gaps and/or less explored arms, which boasts a similar idea as the incentive term in the famous Upper Confidence Bound (UCB) type of algorithms that date back to (Lai and Robbins, 1985; Agrawal, 1995; Auer, 2002), while interestingly the mathematical forms of the two incentive terms are very different.

As the main theoretical result of this paper, we analyze the aggregate regret upper bound of LSA in Theorem 1. We complement the upper bound result with a lower bound theorem (Theorem 20) for any online algorithm. In Remark 2, we compare the upper and lower bounds and show that LSA is instance-wise asymptotically optimal.

We now highlight the technical contributions made in our regret upper bound analysis at a very high level. Please refer to Section 4 for more detailed explanations. In our proof of the upper bound theorem, we first define a global class of events $\{\mathcal{F}_C\}$ (in (14)) which serves as a measurement of how well the arms are explored. Our analysis then goes by two steps. In the first step, we show that $\mathcal{F}_C$ happens with high probability, which intuitively means that all arms are "well explored". In the second step, we show the quantitative upper bound on the mis-classification probability for each arm, when conditioned on $\mathcal{F}_C$. The final regret bound follows by summing up the mis-classification probability for each arm via linearity of expectation. Using this approach, we successfully by-pass the analysis that involves pairs of (or even more) arms, which usually brings in union bound arguments and extra $\ln K$ terms. Indeed, such $\ln K$ slack appears between the upper and lower bounds proved in (Locatelli et al., 2016). In contrast, our LSA algorithm is asymptotically optimal, without any super-constant slack.

Another important technical ingredient that is crucial to the asymptotic optimality analysis is a new concentration inequality for the empirical mean of an arm that uniformly holds over all time periods, which we refer to as the *Variable Confidence Level Bound*. This new inequality helps to reduce an extra $\ln \ln T$ factor in the upper bound. It is also a strict improvement of Hoeffding's celebrated Maximal Inequality, which might be useful in many other problems.

Finally, we highlight that our LSA is *anytime*, i.e., it does not need to know the time horizon $T$ beforehand. LSA does use a universal tuning parameter. However, this parameter does not depend on the instances. As we will show in Section 5, the choice of the parameter is quite robust; and the natural parameter setting leads to superior performance of LSA among a set of very different instances, while APT may suffer from poor performance if the precision parameter is not chosen well for an instance.

**Organization.** The organization of the rest of the paper is as follows. In Section 2 we provide the necessary notation and definitions. Then we present the details of the LSA algorithm in Section 3 and upper bound its aggregate regret in Section 4. In Section 5, we present experiments establishing the empirical advantages of LSA over other algorithms. The instance-wise aggregate regret lower bound theorem is deferred to Appendix E.

## 2 Problem Formulation and Notation

Given an integer $K > 1$, we let $S = [K] \stackrel{\text{def}}{=} \{1, 2, \ldots, K\}$ be the set of $K$ arms in an instance $I$. Each arm $i \in S$ is associated with a distribution $\mathcal{D}_i$ supported on $[0, 1]$ which has an unknown mean $\theta_i$. We are interested in the following dynamic game setting: At any round $t \geq 1$, the learner chooses to pull an arm $i_t$ from $S$ and receives an *i.i.d.* reward sampled from $\mathcal{D}_{i_t}$.

We let $T$, with $T \geq K$, be the *time horizon*, or the *budget* of the game, which is not necessarily known beforehand. We furthermore let $\theta \in (0, 1)$ be the *threshold* of the game. After $T$ rounds, the learner $\mathbb{A}$ has to determine, for every arm $i \in S$, whether or not its mean reward is greater than or equal to $\theta$. So the learner outputs a vector $(d_1, \ldots, d_K) \in \{0, 1\}^K$, where $d_i = 0$ if and only if $\mathbb{A}$ decides that $\theta_i < \theta$. The goal of the Thresholding Bandit Problem (TBP) in this paper is to maximize the expected number of correct labels after $T$ rounds of the game.

More specifically, for any algorithm $\mathbb{A}$, we use $\mathcal{E}_i^{\mathbb{A}}(T)$ to denote the event that $\mathbb{A}$'s decision corresponding to arm $i$ is correct after $T$ rounds of the game. The goal of the TBP algorithm is to minimize the *aggregate regret*, which is the expected number of incorrect classifications for the $K$ arms, i.e.,

$$\mathcal{R}^{\mathbb{A}}(T) = \mathcal{R}^{\mathbb{A}}(I; T) \stackrel{\text{def}}{=} \mathbb{E}\left[\sum_{i=1}^{K} \mathbb{I}_{\{\overline{\mathcal{E}}_i^{\mathbb{A}}(T)\}}\right], \tag{3}$$

where $\overline{\mathcal{E}}$ denotes the complement of event $\mathcal{E}$ and $\mathbb{I}_{\{\text{condition}\}}$ denotes the indicator function.

Let $X_{i,t}$ denote the random variable representing the sample received by pulling arm $i$ for the $t$-th time. We further write

$$\widehat{\theta}_{i,t} \stackrel{\text{def}}{=} \frac{1}{s}\sum_{s=1}^{t} X_{i,s} \quad \text{and} \quad \widehat{\Delta}_{i,t} \stackrel{\text{def}}{=} |\widehat{\theta}_{i,t} - \theta| \tag{4}$$

to denote the *empirical mean* and the *empirical gap* of arm $i$ after being pulled $t$ times, respectively. For a given algorithm $\mathbb{A}$, let $T_i^{\mathbb{A}}(t)$ and $\widehat{\theta}_i^{\mathbb{A}}(t)$ denote the number of times arm $i$ is pulled and the empirical mean reward of arm $i$ after $t$ rounds of the game, respectively. For each arm $i \in S$, we use $\widehat{\Delta}_i^{\mathbb{A}}(t) \stackrel{\text{def}}{=} |\widehat{\theta}_i^{\mathbb{A}}(t) - \theta|$ to denote the empirical gap after $t$ rounds of the game. We will omit the reference to $\mathbb{A}$ when it is clear from the context.

## 3 Our Algorithm

We now motivate our Logarithmic-Sample Algorithm by first designing an optimal but unrealistic algorithm with the assumption that the hardness gaps $\{\Delta_i\}_{i \in S}$ are known beforehand. Now we design the following algorithm $\mathbb{O}$. Suppose the algorithm pulls arm $i$ a total of $x_i$ times and makes a decision based on the empirical mean $\widehat{\theta}_{i,x_i}$: if $\widehat{\theta}_{i,x_i} \geq \theta$, the algorithm decides that $\theta_i \geq \theta$, and decides $\theta_i < \theta$ otherwise. Note that this is all an algorithm can do when the gaps $\Delta_i$ are known. We upper bound the aggregate regret of the algorithm by

$$\mathcal{R}^{\mathbb{O}}(T) = \sum_{i=1}^{K} \mathbb{P}(\overline{\mathcal{E}}_i^{\mathbb{O}}(T)) \leq \sum_{i=1}^{K} \mathbb{P}(|\widehat{\theta}_{i,x_i} - \theta_i| \geq \Delta_i) \leq \sum_{i=1}^{K} 2\exp\left(-2x_i\Delta_i^2\right), \tag{5}$$

where the last inequality follows from Chernoff-Hoeffding Inequality (Proposition 5). Now we would like to minimize the RHS (right-hand-side) of (5), and upper bound the aggregate regret of the optimal algorithm $\mathbb{O}$ by

$$2 \cdot \min_{\substack{x_1+\cdots+x_K=T \\ x_1,\ldots,x_K \in \mathbb{N}}} \sum_{i=1}^{K} \exp(-2x_i\Delta_i^2) = 2\mathscr{P}_2^*(\{\Delta_i\}_{i \in S}, T).$$

Here, for every $c > 0$, we define

$$\mathscr{P}_c^*(\{\Delta_i\}_{i \in S}, T) \stackrel{\text{def}}{=} \min_{\substack{x_1 + \cdots + x_K = T \\ x_1, \ldots, x_K \in \mathbb{N}}} \sum_{i=1}^{K} \exp(-cx_i\Delta_i^2). \tag{6}$$

We naturally introduce the following continuous relaxation of the program $\mathscr{P}_c$, by defining

$$\mathscr{P}_c(\{\Delta_i\}_{i \in S}, T) \stackrel{\text{def}}{=} \min_{\substack{x_1 + \cdots + x_K = T \\ x_1, \ldots, x_K \geq 0}} \sum_{i=1}^{K} \exp(-cx_i\Delta_i^2). \tag{7}$$

$\mathscr{P}_c$ approximates $\mathscr{P}_c^*$ well, as it is straightforward to see that

$$\mathscr{P}_c(\{\Delta_i\}_{i \in S}, T) \leq \mathscr{P}_c^*(\{\Delta_i\}_{i \in S}, T) \leq \mathscr{P}_c(\{\Delta_i\}_{i \in S}, T - K). \tag{8}$$

We apply the Karush-Kuhn-Tucker (KKT) conditions to the optimization problem $\mathscr{P}_2(\{\Delta_i\}_{i \in S}, T)$ and find that the optimal solution satisfies

$$x_i\Delta_i^2 + \ln\Delta_i^{-1} \geq \Phi, \text{ for } i \in S, \tag{9}$$

where $\Phi \stackrel{\text{def}}{=} \max\{x : \sum_{i=1}^{K} \max\{\frac{x - \ln\Delta_i^{-1}}{\Delta_i^2}, 0\} \leq T\}$ is independent of $i \in S$. Furthermore, since $\sum_{i=1}^{K} \max\{\frac{x - \ln\Delta_i^{-1}}{\Delta_i^2}, 0\}$ is an increasing continuous function of $x$, $\Phi$ is indeed well-defined. Please refer to Lemma 10 of Appendix B for the details of the relevant calculations.

In light of (8) and (9), the following algorithm $\mathbb{O}'$ (still, with the unrealistic assumption of the knowledge of the gaps $\{\Delta_i\}_{i \in S}$) incrementally solves $\mathscr{P}_c$ and approximates the algorithm $\mathbb{O}$ – at each time $t$, the algorithm selects the arm $i$ that minimizes $T_i(t-1)\Delta_i^2 + \ln\Delta_i^{-1}$ and plays it.

Our proposed algorithm is very close to $\mathbb{O}'$. Since in reality the algorithm does not have access to the precise gap quantities, we use the empirical estimates $\widehat{\Delta}_i^2$ instead of $\Delta_i^2$ in the $T_i(t-1)\Delta_i^2$ term. For the logarithmic term, if we also use $\ln\widehat{\Delta}_i^{-1}$ instead of $\ln\Delta_i^{-1}$, we may encounter extremely small empirical estimates when the arm is not sufficiently sampled, which would lead to unbounded values of $\ln\widehat{\Delta}_i^{-1}$, and render the arm almost impossible to be sampled in future. To solve this problem, we note that $\mathbb{O}'$ tries to maintain $T_i(t-1)\Delta_i^2$ to be roughly the same across the arms (if ignoring the $\ln\Delta_i^{-1}$ term). In light of this, we use $\sqrt{T_i(t-1)}$ to roughly estimate the order of $\Delta_i^{-1}$. This encourages the exploration of both the arms with larger gaps and the ones with fewer trials.

To summarize, at each time $t$, our algorithm selects the arm $i$ that minimizes $\alpha \cdot T_i(t-1)(\widehat{\Delta}_i(t-1))^2 + 0.5\ln T_i(t-1)$, where $\alpha > 0$ is a universal tuning parameter, and plays the arm. The details of the algorithm are presented in Algorithm 1.

---

**Algorithm 1** Logarithmic-Sample Algorithm, $\text{LSA}(S, \theta)$

---

1: **Input:** A set of arms $S = [K]$, threshold $\theta$
2: **Initialization:** pull each arm once
3: **for** $t = K + 1$ **to** $T$ **do**
4:      Pull arm $i_t = \operatorname*{argmin}_{i \in S} \left( \alpha T_i(t-1)(\widehat{\Delta}_i(t-1))^2 + 0.5\ln T_i(t-1) \right)$
5: **For** each arm $i \in S$, let $d_i \leftarrow 1$ if $\widehat{\theta}_i(T) \geq \theta$ and $d_i \leftarrow 0$ otherwise
6: **Output:** $(d_1, \ldots, d_K)$

---

## 4 Regret Upper Bound for LSA

In this section, we show the upper bound of the aggregate regret of Algorithm 1.

Let $x = \Lambda$ be the solution to the following equation

$$\sum_{i=1}^{K} \left( \mathbb{I}_{\{x \leq \ln\Delta_i^{-1}\}} \cdot \exp(2x) + \mathbb{I}_{\{x > \ln\Delta_i^{-1}\}} \cdot \frac{x - \ln\Delta_i^{-1} + \alpha}{\alpha\Delta_i^2} \right) = \frac{T}{\max\{40/\alpha + 1, 40\}}. \tag{10}$$

Notice that $\sum_{i=1}^{K}(\mathbb{I}_{\{x\leq\ln\Delta_i^{-1}\}}\cdot\exp(2x)+\mathbb{I}_{\{x>\ln\Delta_i^{-1}\}}\cdot\frac{x-\ln\Delta_i^{-1}+\alpha}{\alpha\Delta_i^2})$ is a strictly increasing, continuous function with $x\geq 0$ that becomes $K$ when $x=0$ and goes to infinity when $x\to\infty$. Hence $\Lambda$ is guaranteed to exist and is uniquely defined when $T$ is large. Furthermore, for any $i\in S$, we let

$$\lambda_i \overset{\text{def}}{=} \mathbb{I}_{\{\Lambda\leq\ln\Delta_i^{-1}\}}\cdot\exp(2\Lambda)+\mathbb{I}_{\{\Lambda>\ln\Delta_i^{-1}\}}\cdot\frac{\Lambda-\ln\Delta_i^{-1}+\alpha}{\alpha\Delta_i^2}. \tag{11}$$

We note that $\{\lambda_i\}_{i\in S}$ is the optimal solution to $\mathscr{P}_{2\alpha}(\{\max\{\Delta_i,\exp(-\Lambda)\}\}_{i\in S},T/(\max\{40/\alpha+1,40\}))$. Please refer to Lemma 11 of Appendix B for the detailed calculations.

The goal of this section is to prove the following theorem.

**Theorem 1.** *Let $\mathcal{R}^{\mathrm{LSA}}(T)$ be the aggregate regret incurred by Algorithm 1. When $0<\alpha\leq 8$, and $T\geq\max\{40/\alpha+1,40\}\cdot K$, we have*

$$\mathcal{R}^{\mathrm{LSA}}(T)\leq\Upsilon(\alpha)\cdot\sum_{i\in S}\exp\left(-\frac{\lambda_i\Delta_i^2}{10}\right), \tag{12}$$

*where $\Upsilon(\alpha)=\frac{9.3\cdot\sqrt[8]{2}}{\sqrt[8]{2}-1}\exp\left(\frac{2.1\alpha-\ln\alpha-0.5}{4\alpha}\right)$ is a constant that only depends on the universal tuning parameter $\alpha$.*

**Remark 2.** *If we set $\alpha = 1/20$, then the right-hand side of (12) would be at most $O\left(\sum_{i\in S}\exp\left(-\frac{\lambda_i\Delta_i^2}{10}\right)\right)$. One can verify that*

$$\sum_{i\in S}\exp\left(-\frac{\lambda_i\Delta_i^2}{10}\right)\leq O\left(\mathscr{P}_{1/10}(\{\max\{\Delta_i,\exp(-\Lambda)\}\}_{i\in S},T/801)\right)$$

$$= O\left(\mathscr{P}_{16}(\{\max\{\Delta_i,\exp(-\Lambda)\}\}_{i\in S},T/128160)\right)\leq O\left(\mathscr{P}_{16}(\{\Delta_i\}_{i\in S},T/128160)\right),$$

*where the first inequality is due to Lemma 12 of Appendix B and the equality is because of Lemma 13 of Appendix B. This matches the lower bound stated in Theorem 20 up to constant factors.* [1]

The rest of this section is devoted to the proof of Theorem 1. Before proceeding, we note that the analysis of the APT algorithm (Locatelli et al., 2016) crucially depends on a favorable event stating that the empirical mean of any arm at any time does not deviate too much from the true mean. This requires a union bound that introduces extra factors such as $\ln K$ and $\ln\ln T$. Our analysis adopts a novel approach that does not need a union bound over all arms, and hence avoids the extra $\ln K$ factor. In the second step of our analysis, we introduce the new *Variable Confidence Level Bound* to save the extra doubly logarithmic term in $T$.

Now we dive into details of the proof. Let $B\overset{\text{def}}{=}\{i\in S:\ln\Delta_i^{-1}<\Lambda\}$. Intuitively, $B$ contains the arms that can be well classified by the ideal algorithm $\mathbb{O}$ (described in Section 3), while even the ideal algorithm $\mathbb{O}$ suffers $\Omega(1)$ regret for each arm in $S\setminus B$. In light of this, the key of the proof is to upper bound the regret incurred by the arms in $B$.

Let $\mathcal{R}_B^{\mathrm{LSA}}(T)$ denote the regret incurred by arms in $B$. Note that $\Upsilon(\alpha)\cdot\exp(-\lambda_i\Delta_i^2/10)\geq 1$ for every arm $i\in S\setminus B$, and the regret incurred by each arm is at most 1. Therefore, to establish (12), we only need to show that

$$\mathcal{R}_B^{\mathrm{LSA}}(T)\leq\Upsilon(\alpha)\cdot\sum_{i\in B}\exp\left(\frac{-\lambda_i\Delta_i^2}{10}\right). \tag{13}$$

We set up a few notations to facilitate the proof of (13). We define $\xi_i(t)\overset{\text{def}}{=}\alpha T_i(t)(\widehat{\Delta}_i(t))^2+0.5\ln T_i(t)$ to be the expression inside the $\mathrm{argmin}(\cdot)$ operator in Line 4 of the algorithm, for arm $i$ and at time $t$. We also define $\xi_{i,t}\overset{\text{def}}{=}\alpha t(\widehat{\Delta}_{i,t})^2+0.5\ln t$.

Intuitively, when $\xi_i(t)$ is large, we usually have a larger value for $T_i(t)$, and arm $i$ is better explored. Therefore, $\xi_i(t)$ can be used as a measurement of how well arm $i$ is explored, which directly relates

to the mis-classification probability for classifying the arm. We say that arm $i$ is $C$-*well explored* at time $T$ if there exists $T' \leq T$ such that $\xi_i(T') > C$. For any $C > 0$, we also define the event $\mathcal{F}_C$ to be

$$\mathcal{F}_C \stackrel{\text{def}}{=} \{\exists T' \leq T \; : \; \forall i \in S, \xi_i(T') > C\}. \tag{14}$$

When $\mathcal{F}_C$ happens, we know that all arms are $C$-well explored.

At a higher level, the proof of (13) goes by two steps. First, we show that for $C$ that is almost as large as $\Lambda$, $\mathcal{F}_C$ happens with high probability, which means that every arm is $C$-well explored. Second, we quantitatively relate that being $C$-well explored and the mis-classification probability for classifying each arm, which can be used to further deduce a regret upper bound given the event $\mathcal{F}_C$.

We start by revealing more details about the first step. The following Lemma 3 gives a lower bound on the probability of the event $\mathcal{F}_C$.

**Lemma 3.** $\mathbb{P}(\mathcal{F}_{\Lambda-k}) \geq 1 - \exp(-40k/\alpha)$ *for* $0 \leq k < \Lambda$.

We now introduce the high-level ideas for proving Lemma 3 and defer the formal proofs to Appendix D.2. For any arm $i \in S$ and $C > 0$, let $\tau_{i,C}$ be the random variable representing the smallest positive integer such that $\xi_{i,\tau_{i,C}} > C$ (i.e., $\xi_{i,t} \leq C$ for all $1 \leq t < \tau_{i,C}$). Intuitively, $\tau_{i,C}$ denotes the first time arm $i$ is $C$-well explored. We first show that the distribution of $\tau_{i,C}$ has an exponential tail. Hence, the sum of them with the same $C$ also has an exponential tail. Next, we show that with high probability $\sum_{i=1}^{K} \tau_{i,\Lambda-k} \leq T$ and the probability vanishes exponentially as $k$ increases. In the last step, thanks to the design of the algorithm, we are able to argue that $\sum_{i=1}^{K} \tau_{i,\Lambda-k} \leq T$ implies $\mathcal{F}_{\Lambda-k}$.

We now proceed to the second step of the proof of (13). The following lemma (whose proof is deferred to Appendix D.3) gives an upper bound of regret incurred by arms in $B$ conditioned on $\mathcal{F}_C$.

**Lemma 4.** *If* $k \geq 0.1\alpha$, *then conditioned on* $\mathcal{F}_{\Lambda-k}$,

$$\mathcal{R}_B^{\text{LSA}}(T) \leq \frac{9 \cdot \sqrt[8\alpha]{2}}{\sqrt[8\alpha]{2} - 1} \cdot \sum_{i \in B} \exp\left(-\frac{\lambda_i \Delta_i^2}{10} + \frac{k + \alpha - \ln\alpha - 0.5}{4\alpha}\right).$$

As mentioned before, the key to proving Lemma 4 is to pin down the quantitative relation between the event $\mathcal{F}_C$ and the probability of mis-classifying an arm conditioned on $\mathcal{F}_C$, then the expected regret upper bound can be achieved by summing up the mis-classifying probability for all arms in $B$.

A key technical challenge in our analysis is to design a concentration bound for the empirical mean of an arm (namely arm $i$) that uniformly holds over all time periods. A typical method is to let the length of the confidence band scale linearly with $\sqrt{1/t}$, where $t$ is the number of samples made for the arm. However, this would worsen the failure probability, and lead to an extra $\ln\ln T$ factor in the regret upper bound. To reduce the iterated logarithmic factor, we introduce a novel uniform concentration bound where the ratio between the length of the confidence band and $\sqrt{1/t}$ is almost constant for large $t$, but becomes larger for smaller $t$. Since this ratio is related to the confidence level of the corresponding confidence band, we refer to this new concentration inequality as the *Variable Confidence Level Bound*. More specifically, in Appendix D.3.1, we prove the following lemma.

**Lemma 19** (*Variable Confidence Level Bound*, pre-stated)**.** *Let* $X_1, \ldots, X_L$ *be i.i.d. random variables supported on* $[0, 1]$ *with mean* $\mu$. *For any* $a > 0$ *and* $b > 0$, *it holds that*

$$\mathbb{P}\left(\forall t \in [1, L], \left|\frac{1}{t}\sum_{i=1}^{t} X_i - \mu\right| \leq \sqrt{\frac{a + b\ln(L/t)}{t}}\right) \geq 1 - \frac{2^{b/2+2}}{2^{b/2} - 1}\exp(-a/2).$$

This new inequality greatly helps the analysis of our algorithm, where the intuition is that when conditioned on the event $\mathcal{F}_C$, it is much less likely that fewer number of samples are conducted for arm $i$, and therefore we can afford a less accurate (i.e. bigger) confidence band for its mean value.

It is notable that a similar idea is also adopted in the analysis of the MOSS algorithm (Audibert and Bubeck, 2009) which gives a minimax optimal regret bound for the ordinary multi-armed bandits. However, our Variable Confidence Level Bound is more general. It can replace the usage of Hoeffding's Maximal Inequality in the analysis of MOSS and may find other applications. We

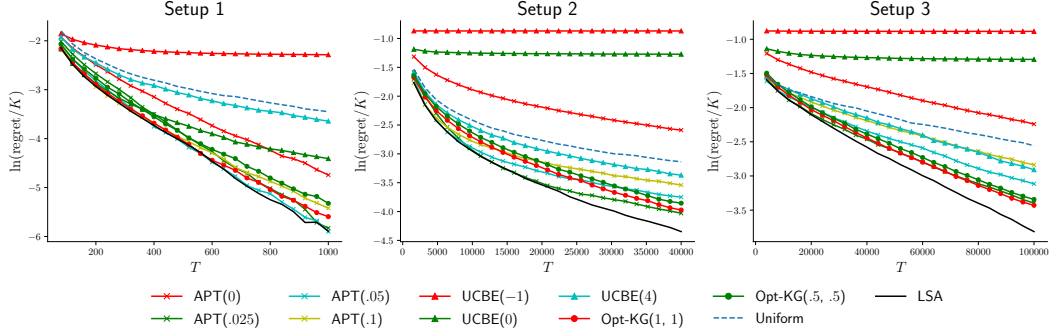

Figure 1: Average aggregate regret on a logarithmic scale for different settings.

additionally remark that in Hoeffding's celebrated Maximal Inequality, the confidence level also changes with time. However, the blow-up factor made to the confidence level in our inequality is only the logarithm of that of Hoeffding's Maximal Inequality. Therefore, if constant factors are ignored, our inequality strictly improves Hoeffding's Maximal Inequality.

The formal proof of Theorem 1 involves a few technical tricks to combine Lemma 3 and Lemma 4 to deduce the final regret bound, and is deferred to Appendix D.1. The lower bound theorem (Theorem 20) that complements Theorem 1 is deferred to Appendix E due to space constraints.

## 5 Experiments

In our experiments, we assume that each arm follows independent Bernoulli distributions with different means. To guarantee a fair comparison, we vary the total number of samples $T$ and compare the empirical average aggregate regret on a logarithmic scale which is averaged over 5000 independent runs. We consider three different choices of $\{\theta_i\}_{i \in S}$:

1. (arithmetic progression I). $K = 10$; $\theta_{1:4} = 0.2 + (0:3) \cdot 0.05, \theta_5 = 0.45, \theta_6 = 0.55$, and $\theta_{7:10} = 0.65 + (0:3) \cdot 0.05$ (see Setup 1 in Figure 1).

2. (arithmetic progression II). $K = 20$; $\theta_{1:20} = 0.405 + (i-1)/100$ (see Setup 2 in Figure 1).

3. (two-group setting). $K = 10$; $\theta_{1:5} = 0.45$, and $\theta_{6:10} = 0.505$ (see Setup 3 in Figure 1).

In our experiments, we fix $\theta = 0.5$. We notice that the choice of $\alpha$ in our LSA is quite robust (see Appendix F.4 for experimental results). To illustrate the performance, we fix $\alpha = 1.35$ in LSA and compare it with four existing algorithms for the TBP problem under a variety of settings. Now we discuss these algorithms and their parameter settings in more details.

- **Uniform:** Given the budget $T$, this method pulls each arm sequentially from 1 to $K$ until budget $T$ is reached such that each arm is sampled roughly $T/K$ times. Then it outputs $\theta_i \geq \theta$ when $\widehat{\theta}_i \geq \theta$.

- **APT($\epsilon$):** Introduced and analyzed in (Locatelli et al., 2016), this algorithm aims to output a set of arms ($\{i \in S : \widehat{\mu}_i \geq \theta\}$) serving as an estimate of the set of arms with means over $\theta + \epsilon$. The natural adaptation of the APT algorithm to our problem corresponds to changing the output: it outputs $\theta_i \geq \theta$ if $\widehat{\theta}_i \geq \theta$ and $\theta_i < \theta$ otherwise. In the experiments, we test the following choices of $\epsilon$: 0, 0.025, 0.05, and 0.1.

- **UCBE($b$):** Introduced and analyzed in (Audibert and Bubeck, 2010), this algorithm aims to identify the best arm (the arm with the largest mean reward). A natural adaptation of this algorithm to TBP is for each time $t$, it pulls $\operatorname{argmin}_{i \in S}(\widehat{\Delta}_i - \sqrt{a/T_i(t-1)})$ where $a$ is a tuning parameter. In (Audibert and Bubeck, 2010), it has been proved optimal when $a = \frac{25}{36}\frac{T-K}{H}$ where $H = \sum_{i \in S} \frac{1}{\Delta_i^2}$. Here we set $a = 4^b \frac{T-K}{H}$ and test three different choices of $b$: $-1$, 0, and 4.

- **Opt-KG($a$, $b$):** Introduced in (Chen et al., 2015), this algorithm also aims to minimize the aggregate regret. It models TBP as a Bayesian Markov decision process where $\{\theta_i\}_{i \in S}$ is

assumed to be drawn from a known Beta prior $\text{Beta}(a, b)$. Here we choose two different priors: $\text{Beta}(1, 1)$ (uniform prior) and $\text{Beta}(0.5, 0.5)$ (Jeffreys prior).

**Comparisons.** In Setup 1, which is a relatively easy setting, LSA works best among all choices of budget $T$. With the right choice of parameter, APT and Opt-KG also achieve satisfactory performance. Though the performance gaps appear to be small, two-tailed paired t-tests of aggregate regrets indicate that LSA is significantly better than most of the other methods, except APT(.05) and APT(.025) (see Table 1 in Appendix F.2).

In Setup 2 and 3, where ambiguous arms close to the threshold $\theta$ are presented, the performance difference between LSA and other methods is more noticeable. LSA consistently outperforms other methods in both settings over almost all choices of budget $T$ with statistical significance. It is worth noting that, though APT works also reasonably well in Setup 2 when $T$ is small, the best parameter $\epsilon$ is different from that for bigger $T$ and other setups. On the other hand, the parameters chosen in LSA are fixed across all setups, indicating that our algorithm is more robust.

We perform additional experiments that due to space limitations are included in Appendix F.3. In all setups, LSA outperforms its competitors with various parameter choices.

## 6  Conclusion

In this paper we introduce an algorithm that minimizes the aggregate regret for the thresholding bandit problem. Our algorithm LSA makes use of a novel approach inspired by the optimal allocation scheme of the budget when the reward gaps are known ahead of time. When compared to APT, LSA uses an additional term, similar in spirit to the UCB-type algorithms though mathematically different, that encourages the exploration of arms that have bigger gaps, and/or those have not been sufficiently explored. Moreover, LSA is anytime and robust, while the precision parameter $\epsilon$ needed in the APT algorithm is highly sensitive and hard to choose. Besides showing empirically that LSA performs better than APT for different values of $\epsilon$ and other algorithms in a variety of settings, we also employ novel proof ideas that eliminate the logarithmic terms usually brought in by the straightforward union bound argument, design the new *Variable Confidence Level Bound* that strictly improves Hoeffding's celebrated Maximal Inequality, and prove that LSA achieves instance-wise asymptotically optimal aggregate regret.

## Footnotes

[1]While the constants may seem large, we emphasize that i) we make no effort in optimizing the constants in asymptotic bounds, ii) most of the constants come from the lower bound, while the constant factor in our upper bound is 10, and iii) we believe that the actual constant of our algorithm is quite small, as the experimental evaluation in the later section demonstrates that our algorithm performs very well in practice.

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
