[Supplementary Material]

# Appendix

## A  Probability Tools

**Proposition 5** (Chernoff-Hoeffding Inequality (Hoeffding, 1963))**.** *Let $\{X_i\}_{i\in[t]}$ be a list of independent random variables supported on $[0,1]$ and set $X = \frac{1}{t}\sum_{i=1}^{t} X_i$. Then, for every $\epsilon > 0$, it holds that*

$$\mathbb{P}(|X - \mathbb{E}[X]| \geq \epsilon) \leq 2\exp(-2t\epsilon^2).$$

**Proposition 6** (Restatement of Theorem 5.1(ii) in (Janson, 2018))**.** *Let $\{X_i\}_{i\in[t]}$ be a list of independent random variables such that $\mathbb{P}(X_i > x) \leq \exp(-a_i x)$ for $x > 0$. And let $\mu = \sum_{i=1}^{t} \frac{1}{a_i}$. Then for any $\lambda \geq 1$, it holds that*

$$\mathbb{P}(X \geq \lambda\mu) \leq \exp(1 - \lambda).$$

**Proposition 7** (Hoeffding's Maximal Inequality (Hoeffding, 1963))**.** *Let $\{X_i\}_{i\in[t]}$ be a list of i.i.d. random variables supported on $[0,1]$ and set $\mu = \mathbb{E}[X_1]$. Then, for any $\epsilon > 0$, it holds that*

$$\mathbb{P}(\forall i \in [t], X_1 + X_2 + \cdots + X_i \geq i\mu + \epsilon) \leq \exp\left(-\frac{2\epsilon^2}{t}\right).$$

**Proposition 8** (Restatement of Lemma 2.6 in (Tsybakov, 2009))**.** *Let $P$ and $Q$ be two probability distributions supported on some set $\mathcal{X}$. Then for every set $A \subset \mathcal{X}$, one has*

$$\mathbb{P}_{X\sim P}(A) + \mathbb{P}_{X\sim Q}(\overline{A}) \geq \frac{1}{2}\exp(-D_{\mathrm{KL}}(P \parallel Q)),$$

*where $\overline{A}$ denotes the complement of $A$ and $D_{\mathrm{KL}}$ denotes the Kullback-Leibler divergence between $P$ and $Q$ given by*

$$D_{\mathrm{KL}}(P \parallel Q) \overset{\text{def}}{=} \sum_{x\in\mathcal{X}} P(x)\ln\left(\frac{P(x)}{Q(x)}\right).$$

**Proposition 9** (Restatement of Lemma 15.1 in (Lattimore and Szepesvári, 2018))**.** *Let $v = P_1 \otimes \cdots \otimes P_K$ and $v' = P_1' \otimes \cdots \otimes P_K'$ be the reward distributions of two $K$-armed bandits. Assuming $D_{\mathrm{KL}}(P_i, P_i') < +\infty$ for any arm $i \in [K]$. Fix some policy $\pi$ and let $\mathbb{P}_v = \mathbb{P}_{v\pi}$ and $\mathbb{P}_v = \mathbb{P}_{v'\pi}$ be the two probability measures induced by the $n$-round interconnection of $\pi$ and $v$ (respectively, $\pi$ and $v'$). Then*

$$D_{\mathrm{KL}}(\mathbb{P}_v \parallel \mathbb{P}_{v'}) = \sum_{i=1}^{K} \mathbb{E}_v[T_i(n)] \cdot D_{\mathrm{KL}}(P_i \parallel P_i'),$$

*where $T_i(n)$ is the random variable denoting the number of times arm $i$ is pulled.*

## B  Properties of $\mathscr{P}_c$

We first show the optimal solution to $\mathscr{P}_c(\{\Delta_i\}_{i\in S}, T)$ by proving the following lemma.

**Lemma 10.** *If $c > 0$, then the optimal solution to $\mathscr{P}_c(\{\Delta_i\}_{i\in S}, T)$ can be expressed in the following form*

$$x_i = \max\left\{\frac{\Phi_c - \ln\Delta_i^{-1}}{c\Delta_i^2/2}, 0\right\},$$

*where $\Phi_c \overset{\text{def}}{=} \max\{x : \sum_{i=1}^{K}\max\{\frac{x-\ln\Delta_i^{-1}}{c\Delta_i^2/2}, 0\} \leq T\}$.*

*Proof.* Since $\sum_{i=1}^{K}\max\{\frac{x-\ln\Delta_i^{-1}}{c\Delta_i^2/2}, 0\}$ is an increasing continuous function on $x$, $\Phi_c$ is indeed well-defined.

We apply KKT conditions (see Proposition 8.7.2 in (Matouek and Gärtner, 2006)) to solve the minimization problem $\mathscr{P}_c(\{\Delta_i\}_{i\in S}, T)$. Concretely, the KKT conditions apply to $\mathscr{P}_c(\{\Delta_i\}_{i\in S}, T)$ gives

$$(-c\Delta_i^2)\exp(-cx_i\Delta_i^2) - u_i + v = 0 \text{ for } i \in [K]$$

$$u_i x_i = 0 \text{ for } i \in [K]$$
$$u_i \leq 0 \text{ for } i \in [K]$$
$$x_i \geq 0 \text{ for } i \in [K]$$
$$\sum_{i=1}^{K} x_i = T,$$

where $u_i$ for $i \in [K]$ and $v$ are $K + 1$ newly-introduced variables. In particular, if $x_i > 0$, then $u_i = 0$ and it holds that

$$\frac{c}{2} x_i \Delta_i^2 + \ln \Delta_i^{-1} = \frac{1}{2} \ln \frac{c}{v}. \tag{15}$$

It is easy to see the solution $x_i = \max\{\frac{\Phi_c - \ln \Delta_i^{-1}}{c\Delta_i^2/2}, 0\}$ for $i \in [K]$ satisfies (15) and is a minimum point. $\qquad\square$

For any positive number $c > 0$, let $x = \Psi_c$ be the solution to

$$\sum_{i=1}^{K} \left( \mathbb{I}_{\{x \leq \ln \Delta_i^{-1}\}} \cdot \exp(2x) + \mathbb{I}_{\{x > \ln \Delta_i^{-1}\}} \cdot \frac{x - \ln \Delta_i^{-1} + c/2}{c\Delta_i^2/2} \right) = T.$$

Note that

$$\sum_{i=1}^{K} \left( \mathbb{I}_{\{x \leq \ln \Delta_i^{-1}\}} \cdot \exp(2x) + \mathbb{I}_{\{x > \ln \Delta_i^{-1}\}} \cdot \frac{x - \ln \Delta_i^{-1} + c/2}{c\Delta_i^2/2} \right)$$

is a strictly increasing continuous function on $x$ that equals $K$ when $x = 0$ and tends to infinity when $x \to \infty$. Hence $\Psi_c$ exists and is uniquely defined.

Then we derive the optimal solution to $\mathscr{P}_c(\{\max\{\Delta_i, \exp(-\Psi_c)\}\}_{i \in S}, T)$, as follows.

**Lemma 11.** *If $c > 0$, then the optimal solution to $\mathscr{P}_c(\{\max\{\Delta_i, \exp(-\Psi_c)\}\}_{i \in S}, T)$ can be expressed in the following form*

$$x_i = \mathbb{I}_{\{\Psi_c \leq \ln \Delta_i^{-1}\}} \cdot \exp(2\Psi_c) + \mathbb{I}_{\{\Psi_c > \ln \Delta_i^{-1}\}} \cdot \frac{\Psi_c - \ln \Delta_i^{-1} + c/2}{c\Delta_i^2/2}.$$

*Proof.* By Lemma 10, the optimal solution to $\mathscr{P}_c(\{\max\{\Delta_i, \exp(-\Psi_c)\}\}_{i \in S}, T)$ can be expressed as

$$\frac{c}{2} x_i \max\{\Delta_i, \exp(-\Psi_c)\}^2 + \ln \max\{\Delta_i, \exp(-\Psi_c)\}^{-1} = \Phi_c,$$

where

$$\Phi_c = \max \left\{ x : \sum_{i=1}^{K} \max \left\{ \frac{x - \ln \max\{\Delta_i, \exp(-\Psi_c)\}^{-1}}{c \max\{\Delta_i, \exp(-\Psi_c)\}^2/2}, 0 \right\} \leq T \right\}.$$

It is easy to see that $\Phi_c = \Psi_c + c/2$. Therefore the optimal solution to $\mathscr{P}_c(\{\max\{\Delta_i, \exp(-\Psi_c)\}\}_{i \in S}, T)$ is

$$x_i = \max \left\{ \frac{\Phi_c - \ln \max\{\Delta_i, \exp(-\Psi_c)\}^{-1}}{\frac{c}{2} \max\{\Delta_i, \exp(-\Psi_c)\}^2}, 0 \right\}$$

$$= \mathbb{I}_{\{\Psi_c \leq \ln \Delta_i^{-1}\}} \cdot \exp(2\Psi_c) + \mathbb{I}_{\{\Psi_c > \ln \Delta_i^{-1}\}} \cdot \frac{\Psi_c - \ln \Delta_i^{-1} + c/2}{c\Delta_i^2/2},$$

proving this lemma. $\qquad\square$

Using Lemma 11, we derive the following useful inequality.

**Lemma 12.** *Suppose $c > 0$ and let $\{x_i^*\}_{i \in S}$ be the solution to $\mathscr{P}_c(\{\max\{\Delta_i, \exp(-\Psi_c)\}\}_{i \in S}, T)$. Then*

$$\sum_{i \in S} \exp(-cx_i^* \Delta_i^2) \leq \exp(c) \mathscr{P}_c(\{\max\{\Delta_i, \exp(-\Psi_c)\}\}_{i \in S}, T).$$

*Proof.* By Lemma 11, the optimal solution to $\mathscr{P}_c(\{\max\{\Delta_i, \exp(-\Psi_c)\}\}_{i\in S}, T)$ can be expressed as

$$x_i^* = \mathbb{I}_{\{\Psi_c \leq \ln \Delta_i^{-1}\}} \cdot \exp(2\Psi_c) + \mathbb{I}_{\{\Psi_c > \ln \Delta_i^{-1}\}} \cdot \frac{\Psi_c - \ln \Delta_i^{-1} + c/2}{c\Delta_i^2/2}. \tag{16}$$

Therefore, we obtain

$$\sum_{i\in S} \exp\left(-cx_i^*\Delta_i^2\right)$$
$$\leq \exp(c) \sum_{i\in S} \exp\left(-cx_i^* \max\{\Delta_i, \exp(-\Phi_c)\}^2\right)$$
$$= \exp(c)\mathscr{P}_c(\{\max\{\Delta_i, \exp(-\Psi_c)\}\}_{i\in S}, T),$$

and this lemma follows. $\qquad\square$

Finally, we will show how the value of $\mathscr{P}_c(\{\Delta_i\}_{i\in S}, T)$ will change when $c$ is changed.

**Lemma 13.** *If $c, c' > 0$, then*

$$\mathscr{P}_c(\{\Delta_i\}_{i\in S}, T) = \mathscr{P}_{c'}(\{\Delta_i\}_{i\in S}, Tc/c').$$

*Proof.* We observe that for any sequence of positive numbers $\{x_i\}_{i\in S}$,

$$\sum_{i=1}^{K} \exp(-cx_i\Delta_i^2) = \sum_{i=1}^{K} \exp(-c' \cdot (cx_i/c')\Delta_i^2).$$

Suppose $\{x_i\}_{i\in S}$ is the optimal solution to $\mathscr{P}_c(\{\Delta_i\}_{i\in S}, T)$. Then $\{cx_i/c'\}_{i\in S}$ is a feasible solution to $\mathscr{P}_{c'}(\{\Delta_i\}_{i\in S}, Tc/c')$. Hence we obtain $\mathscr{P}_{c'}(\{\Delta_i\}_{i\in S}, Tc/c') \leq \mathscr{P}_c(\{\Delta_i\}_{i\in S}, T)$. On the other hand, using a similar argument, we can also obtain $\mathscr{P}_c(\{\Delta_i\}_{i\in S}, T) \leq \mathscr{P}_{c'}(\{\Delta_i\}_{i\in S}, Tc/c')$. Therefore, it holds that

$$\mathscr{P}_c(\{\Delta_i\}_{i\in S}, T) = \mathscr{P}_{c'}(\{\Delta_i\}_{i\in S}, Tc/c'),$$

and the lemma follows. $\qquad\square$

## C   Hard Instances for the Uniform Sampling Approach

In this section, we describe a class of bad instances for the uniform sampling approach. In such instances, we show that, to achieve the same order of regret, the uniform sampling approach needs at least $\Omega(K)$ times more budget than the optimal policy.

We fix the threshold $\theta = 0.5$. For each $K \geq 20$, we construct two instances $I_1$ and $I_2$. In $I_1$, we set $\theta_1 = 0.5 - \sqrt{1/(K-1)}$, and $\theta_{2:K} = 0.5 + \sqrt{0.1}$. In $I_2$, we set $\theta_1 = 0.5 + \sqrt{1/(K-1)}$, and $\theta_{2:K} = 0.5 + \sqrt{0.1}$. Hence for both instances, $\Delta_1 = \sqrt{1/(K-1)}$ and $\Delta_{2:K} = \sqrt{0.1}$. Suppose $T = 2K(K-1)t_0$ where $t_0 \geq 10$. For simplicity, we use $\mathcal{R}^{\mathrm{uni}}(I; T)$ and $\mathcal{R}^{\mathrm{opt}}(I; T)$ to represent the regret incurred by the uniform sampling approach and the optimal policy on instance $I$ respectively.

We now bound $\mathcal{R}^{\mathrm{uni}}(I; T)$ and $\mathcal{R}^{\mathrm{opt}}(I; T)$ in sequence. We first consider the uniform sampling approach and give a lower bound of regret incurred by it. Note that given $T = 2K(K-1)t_0$, the uniform sampling approach will play each arm $2(K-1)t_0$ times. Let $\mathcal{K}$ denote the event that the classification for arm 1 is incorrect. Define $\mathbb{P}_I[\cdot]$ as the probability induced by performing the uniform sampling approach on instance $I$. We have

$$\max\{\mathcal{R}^{\mathrm{uni}}(I_1; T), \mathcal{R}^{\mathrm{uni}}(I_2; T)\} \geq \max\{\mathbb{P}_{I_1}(\mathcal{K}), \mathbb{P}_{I_2}(\mathcal{K})\}$$
$$\geq \frac{1}{2} \exp\left(-16 \cdot \left(\sqrt{1/(K-1)}\right)^2 \cdot 2(K-1)t_0\right)$$
$$= \Omega(\exp(-32t_0)) \tag{17}$$

where the second inequality is obtained by applying Theorem 20 when there is only one arm.

Next we derive an upper bound of regret incurred by the optimal policy. By setting $x_1 = K(K - 1)t_0 - (K-1) \ln K$ and $x_{2:K} = Kt_0 + \ln K$, and using Chernoff-Hoeffding Inequality (Proposition 5), we have for any instance $I \in \{I_1, I_2\}$, it holds that

$$
\begin{aligned}
&\mathcal{R}^{\mathrm{opt}}(I; T) \\
&\leq 2 \exp\left(-2 \cdot \frac{1}{K-1} \cdot (K(K-1)t_0 - (K-1)\ln K)\right) + 2(K-1)\exp(-2 \cdot 0.1 \cdot (Kt_0 + \ln K)) \\
&\leq 2(K+1)^2 \exp(-0.2Kt_0)
\end{aligned}
\tag{18}
$$

For any $\epsilon \leq 1/(K+1)$, according to (17), there exists an instance $I' \in \{I_1, I_2\}$ such that, to achieve $\epsilon$ regret, the uniform sampling approach needs at least $\Omega(K^2 \ln \epsilon^{-1})$ budget. However, by (18), the optimal policy only needs at most $10(K-1)(\ln \frac{2}{\epsilon} + 2\ln(K+1)) \leq 20(K-1)\ln\frac{2}{\epsilon} = O(K\ln\epsilon^{-1})$ plays for $I'$.

# D   Missing Proofs in Section 4

## D.1   Proof of Theorem 1

For convenience, we define the real-valued function $f(x) \stackrel{\text{def}}{=} \alpha x + \ln \alpha + 0.5 - \alpha$ and use $f^{-1}$ to denote its inverse. Also, we use $\mathcal{R}_B^{\mathrm{LSA}}(T)_{\mid \mathcal{F}}$ to denote the regret incurred by arms in $B$ when conditioned on event $\mathcal{F}$.

*Proof of Theorem 1.* As discussed before, we only need to establish (13), i.e.,

$$
\mathcal{R}_B^{\mathrm{LSA}}(T) \leq \Upsilon(\alpha) \cdot \sum_{i \in B} \exp\left(\frac{-\lambda_i \Delta_i^2}{10}\right).
$$

Let $\Lambda' = \alpha \lfloor \frac{\Lambda}{\alpha} - 0.1 \rfloor$ and define the events $\mathcal{G}_0 \stackrel{\text{def}}{=} \mathcal{F}_{\Lambda'}$, $\mathcal{G}_k \stackrel{\text{def}}{=} \bigwedge_{i=0}^{k-1}\overline{\mathcal{F}}_{\Lambda'-\alpha i} \wedge \mathcal{F}_{\Lambda'-\alpha k}$ if $1 \leq k \leq \lfloor \frac{\Lambda}{\alpha} - 0.1 \rfloor$, and $\mathcal{G}_{\lfloor \frac{\Lambda}{\alpha}-0.1 \rfloor + 1} \stackrel{\text{def}}{=} \bigwedge_{i=0}^{\lfloor \frac{\Lambda}{\alpha}-0.1 \rfloor}\overline{\mathcal{F}}_{\alpha i}$. Note that the events $\mathcal{G}_0, \ldots, \mathcal{G}_{\lfloor \frac{\Lambda}{\alpha}-0.1 \rfloor+1}$ form a partition of the total probability space. Then,

$$
\begin{aligned}
\mathcal{R}_B^{\mathrm{LSA}}(T) &= \sum_{k=0}^{\lfloor \frac{\Lambda}{\alpha}-0.1 \rfloor + 1} \mathcal{R}_B^{\mathrm{LSA}}(T)_{\mid \mathcal{G}_k} \cdot \mathbb{P}(\mathcal{G}_k) \\
&\leq \mathcal{R}_B^{\mathrm{LSA}}(T)_{\mid \mathcal{F}_{\Lambda'}} + \sum_{k=1}^{\lfloor \frac{\Lambda}{\alpha}-0.1 \rfloor} \mathcal{R}_B^{\mathrm{LSA}}(T)_{\mid \mathcal{F}_{\Lambda'-\alpha k}} \cdot \mathbb{P}(\overline{\mathcal{F}}_{\Lambda'-\alpha(k-1)}) + |B| \mathbb{P}(\overline{\mathcal{F}}_0).
\end{aligned}
\tag{19}
$$

Notice that that $\xi_{i,10} \geq 0.5 \ln 10 > 0$. Moreover, since $T \geq 10K$, after $T$ rounds it holds that $\xi_i(T) > 0$ for all $i \in S$. Therefore, $\mathbb{P}(\overline{\mathcal{F}}_0) = 0$. Recall that $f^{-1}(y) = \frac{y+\alpha-\ln\alpha-0.5}{\alpha}$. Combining Lemma 3 and Lemma 4, we upper bound (19) by

$$
\begin{aligned}
&\sum_{i \in B} \frac{9 \cdot \sqrt[8\alpha]{2}}{\sqrt[8\alpha]{2}-1} \exp\left(-\frac{\lambda_i \Delta_i^2}{10} + f^{-1}(1.1\alpha)/4\right) \\
&\quad + \sum_{k=1}^{\lfloor \frac{\Lambda}{\alpha}-0.1 \rfloor} \sum_{i \in B} \frac{9 \cdot \sqrt[8\alpha]{2}}{\sqrt[8\alpha]{2}-1} \exp\left(-\frac{\lambda_i \Delta_i^2}{10} + f^{-1}((k+1.1)\alpha)/4\right)\exp(-40(k-0.9)) \\
&\leq \frac{9 \cdot \sqrt[8\alpha]{2}}{\sqrt[8\alpha]{2}-1} \exp\left(f^{-1}(1.1\alpha)/4\right)\left(1 + \sum_{k=1}^{\lfloor \frac{\Lambda}{\alpha}-0.1 \rfloor} \exp(-39.75k+36)\right) \cdot \sum_{i \in B} \exp\left(-\frac{\lambda_i \Delta_i^2}{10}\right) \\
&\leq \frac{9.3 \cdot \sqrt[8\alpha]{2}}{\sqrt[8\alpha]{2}-1} \exp\left(\frac{2.1\alpha - \ln\alpha - 0.5}{4\alpha}\right) \cdot \sum_{i \in B} \exp\left(-\frac{\lambda_i \Delta_i^2}{10}\right)
\end{aligned}
$$

This completes the proof of (13). $\qquad\square$

## D.2 Proof of Lemma 3

The goal of this subsection is to establish the following lemma which gives a lower bound on the probability of $\mathcal{F}_C$.

**Lemma 3 (restated).** $\mathbb{P}(\mathcal{F}_{\Lambda-k}) \geq 1 - \exp(-40k/\alpha)$ *for* $0 \leq k < \Lambda$.

To prove Lemma 3, we make use of Lemma 14 and Lemma 15, and defer their proofs to the later part of this subsection.

Recall that for any arm $i \in S$ and $C > 0$, $\tau_{i,C}$ is the random variable representing the smallest positive integer such that $\xi_{i,\tau_{i,C}} > C$. The following Lemma 14 shows an exponentially small tail of the distribution of $\tau_{i,C}$.

**Lemma 14.** *For any arm* $i \in S$, *and* $C > 0$, *we have the following statements:*

(a) $\tau_{i,C} \leq 2 \exp(2C)$;
(b) *if* $C > \ln \Delta_i^{-1}$, *then for any* $k \geq 1$, $\tau_{i,C}$ *satisfies*

$$\mathbb{P}\left(\tau_{i,C} > \frac{40}{\alpha} \cdot \frac{C - \ln \Delta_i^{-1} + k}{\Delta_i^2}\right) \leq 2 \exp(-40k/\alpha).$$

Based on Lemma 14, we are able to show that $\sum_{i=1}^K \tau_{i,C}$ also follows an exponential distribution, which leads to the following lemma.

**Lemma 15.** $\mathbb{P}(\sum_{i=1}^K \tau_{i,\Lambda-k} \leq T) \geq 1 - \exp(-40k/\alpha)$ *for all* $0 \leq k < \Lambda$.

We are now ready to prove the main lemma (Lemma 3) of this subsection.

*Proof of Lemma 3.* By Lemma 15, it suffices to prove that $\mathcal{F}_{\Lambda-k}$ occurs when $\sum_{i=1}^K \tau_{i,\Lambda-k} \leq T$. So we assume that all the random rewards are generated before the algorithm starts and that $\sum_{i=1}^K \tau_{i,\Lambda-k} \leq T$.

Since $\xi_{i,t} \geq \ln t$, it is easy to see that there exists $T^*$ satisfying $\max_{i \in S} \xi_i(T^*) > \Lambda - k$ and $\max_{i \in S} \xi_i(t) \leq \Lambda - k$ for any $1 \leq t < T^*$. We claim that $T^* \leq T$. Indeed, notice that for any arm $i \in S$ and $t \leq T^* - 1$, $\xi_i(t) \leq \Lambda - k$. Hence $T_i(T^* - 1) < \tau_{i,\Lambda-k}$, and so $T^* - 1 = \sum_{i=1}^K T_i(T^* - 1) < \sum_{i=1}^K \tau_{i,\Lambda-k} \leq T$. Therefore $T^* \leq T$.

Now, we assume without loss of generality that for arm $i^* \in S$, $\xi_{i^*}(T^*) > \Lambda - k$. Since at time $t$ Algorithm 1 pulls $\operatorname{argmin}_{i \in S} \xi_i(t-1)$, arm $i^*$ will not be pulled until all the other arms $i \in S \setminus \{i^*\}$ satisfy $\xi_i(t-1) > \Lambda - k$. Since $\sum_{i=1}^K \tau_{i,\Lambda-k} \leq T$, then we can find $T^\natural$ such that $T^* \leq T^\natural \leq T$ and $\xi_i(T^\natural) > \Lambda - k$ for any arm $i \in S$. This proves the lemma. $\square$

### D.2.1 Proof of Lemma 14

**Lemma 14 (restated).** *For any arm* $i \in S$, *and* $C > 0$, *we have the following statements:*

(a) $\tau_{i,C} \leq 2 \exp(2C)$;
(b) *if* $C > \ln \Delta_i^{-1}$, *then for any* $k \geq 1$, $\tau_{i,C}$ *satisfies*

$$\mathbb{P}\left(\tau_{i,C} > \frac{40}{\alpha} \cdot \frac{C - \ln \Delta_i^{-1} + k}{\Delta_i^2}\right) \leq 2 \exp(-40k/\alpha).$$

*Proof.* We first prove Lemma 14(a). Note that if $t \geq \lfloor 2 \exp(2C) \rfloor$, then we have $\xi_{i,t} > 0.5 \ln t \geq C$. Hence $t \leq \lfloor 2 \exp(2C) \rfloor \leq 2 \exp(2C)$ as desired.

Now we prove Lemma 14(b). Note that $\forall k \geq 1$,

$$\mathbb{P}\left(\tau_{i,C} > \frac{40}{\alpha} \cdot \frac{C - \ln \Delta_i^{-1} + k}{\Delta_i^2}\right) \leq \mathbb{P}\left(\xi_{i,\tau_{i,C}} \leq C \mid \tau_{i,C} = \left\lfloor \frac{40}{\alpha} \cdot \frac{C - \ln \Delta_i^{-1} + k}{\Delta_i^2} \right\rfloor\right).$$

Assuming $\tau_{i,C} = \left\lfloor \frac{40}{\alpha} \cdot \frac{C - \ln \Delta_i^{-1} + k}{\Delta_i^2} \right\rfloor$ and $|\widehat{\Delta}_{i,\tau_{i,C}} - \Delta_i| < \sqrt{10}\Delta_i/4$, we get that

$$
\begin{aligned}
\xi_{i,\tau_{i,C}} &= \alpha \tau_{i,C} (\widehat{\Delta}_{i,\tau_{i,C}})^2 + 0.5 \ln \tau_{i,C} \\
&> \alpha \cdot \left\lfloor \frac{40}{\alpha} \cdot \frac{C - \ln \Delta_i^{-1} + k}{\Delta_i^2} \right\rfloor \cdot ((1 - \sqrt{10}/4)\Delta_i)^2 + 0.5 \ln \tau_{i,C} \\
&\geq \alpha \cdot \frac{4}{5} \cdot \frac{40}{\alpha} \cdot \frac{C - \ln \Delta_i^{-1} + k}{\Delta_i^2} \cdot ((1 - \sqrt{10}/4)\Delta_i)^2 + 0.5 \ln \tau_{i,C} \\
&> C - \ln \Delta_i^{-1} + k + 0.5 \ln \tau_{i,C} > C,
\end{aligned}
$$

where we used $\tau_{i,C} = \left\lfloor \frac{40}{\alpha} \cdot \frac{C - \ln \Delta_i^{-1} + k}{\Delta_i^2} \right\rfloor \geq \frac{4}{5} \cdot \frac{40}{\alpha} \cdot \frac{C - \ln \Delta_i^{-1} + k}{\Delta_i^2} > \frac{4}{\Delta_i^2}$ when $\alpha \leq 8$ and $k \geq 1$.

Therefore, we have that

$$
\begin{aligned}
\mathbb{P} &\left( \xi_{i,\tau_{i,C}} \leq C \mid \tau_{i,C} = \left\lfloor \frac{40}{\alpha} \cdot \frac{C - \ln \Delta_i^{-1} + k}{\Delta_i^2} \right\rfloor \right) \\
&\leq \mathbb{P} \left( |\widehat{\Delta}_{i,\tau_{i,C}} - \Delta_i| \geq \frac{\sqrt{10}}{4}\Delta_i \mid \tau_{i,C} = \left\lfloor \frac{40}{\alpha} \cdot \frac{C - \ln \Delta_i^{-1} + k}{\Delta_i^2} \right\rfloor \right) \\
&\leq \mathbb{P} \left( |\widehat{\theta}_{i,\tau_{i,C}} - \theta_i| \geq \frac{\sqrt{10}}{4}\Delta_i \mid \tau_{i,C} = \left\lfloor \frac{40}{\alpha} \cdot \frac{C - \ln \Delta_i^{-1} + k}{\Delta_i^2} \right\rfloor \right) \\
&\leq 2 \exp \left( -2 \cdot \frac{4}{5} \cdot \frac{40}{\alpha} \cdot \frac{C - \ln \Delta_i^{-1} + k}{\Delta_i^2} \cdot (\sqrt{10}\Delta_i/4)^2 \right) \\
&\leq 2 \exp(-40k/\alpha),
\end{aligned}
$$

where the second inequality follows since $|\widehat{\Delta}_{i,\tau_{i,C}} - \Delta_i| = ||\widehat{\theta}_{i,\tau_{i,C}} - \tau| - |\theta_i - \tau|| \leq |\widehat{\theta}_{i,\tau_{i,C}} - \theta_i|$, and the third inequality follows from Chernoff-Hoeffding Inequality (Proposition 5). This proves the desired result. □

### D.2.2 Proof of Lemma 15

**Lemma 15 (restated).** $\mathbb{P}(\sum_{i=1}^{K} \tau_{i,\Lambda-k} \leq T) \geq 1 - \exp(-40k/\alpha)$ for all $0 \leq k < \Lambda$.

*Proof.* Define the set $A \overset{\text{def}}{=} \{i \in S : \Lambda > \ln \Delta_i^{-1} + k\}$. We can assume without loss of generality that $A$ is not empty. Let $\mathcal{E}_1$ be the event

$$
\sum_{i \in S \setminus A} \tau_{i,\Lambda-k} \leq \sum_{i \in S \setminus A} \left( \mathbb{I}_{\{\Lambda \leq \ln \Delta_i^{-1}\}} \cdot 2 \exp(2\Lambda) + \mathbb{I}_{\{\Lambda > \ln \Delta_i^{-1}\}} \cdot \frac{40}{\alpha} \cdot \frac{\Lambda - \ln \Delta_i^{-1} + 1 + \alpha/40}{\Delta_i^2} \right);
\tag{20}
$$

and let $\mathcal{E}_2$ be the event

$$
\sum_{i \in A} \tau_{i,\Lambda-k} \leq \sum_{i \in A} \frac{40}{\alpha} \cdot \frac{\Lambda - \ln \Delta_i^{-1} + 1 + \alpha/40}{\Delta_i^2}.
\tag{21}
$$

Note that when $\mathcal{E}_1$ and $\mathcal{E}_2$ hold, we have $\sum_{i=1}^{K} \tau_{i,\Lambda-k} \leq \sum_{i=1}^{K} \max\{40/\alpha + 1, 40\}\lambda_i = T$. Hence $\mathbb{P}(\sum_{i=1}^{K} \tau_{i,\Lambda-k} > T) \leq \mathbb{P}(\overline{\mathcal{E}}_1) + \mathbb{P}(\overline{\mathcal{E}}_2)$, and since $\mathcal{E}_1$ always holds by Lemma 14(a), we have

$$
\mathbb{P} \left( \sum_{i=1}^{K} \tau_{i,\Lambda-k} > T \right) \leq \mathbb{P} \left( \sum_{i \in A} \tau_{i,\Lambda-k} > \sum_{i \in A} \frac{40}{\alpha} \cdot \frac{\Lambda - \ln \Delta_i^{-1} + 1 + \alpha/40}{\Delta_i^2} \right).
$$

Now, for any arm $i \in A$, let $z_i = \tau_{i,\Lambda-k} - \frac{40}{\alpha} \cdot \frac{\Lambda - k - \ln \Delta_i^{-1} + 1}{\Delta_i^2}$. By Lemma 14(b), for any $x \geq 0$, $z_i$ satisfies

$$
\mathbb{P}(z_i > x) \leq 2 \exp \left( -\frac{40}{\alpha}(\alpha x \Delta_i^2/40 + 1) \right) \leq \exp(-x\Delta_i^2).
$$

Applying Proposition 6, we have that for any $\lambda \geq 1$,

$$\mathbb{P}\left(\sum_{i \in A} z_i > \lambda \cdot \sum_{i \in A} \frac{1}{\Delta_i^2}\right) \leq \exp(1 - \lambda).$$

Therefore,

$$\mathbb{P}\left(\sum_{i \in A} \tau_{i, \Lambda - k} > \sum_{i \in A} \frac{40}{\alpha} \cdot \frac{\Lambda - \ln \Delta_i^{-1} + 1 + \alpha/40}{\Delta_i^2}\right)$$

$$= \mathbb{P}\left(\sum_{i \in A} z_i > (40k/\alpha + 1) \cdot \sum_{i \in A} \frac{1}{\Delta_i^2}\right) \leq \exp(-40k/\alpha).$$

This completes the proof of the lemma. $\qquad\square$

### D.3 Proof of Lemma 4

Recall that we defined $B = \{i \in S \; : \; \Lambda > \Delta_i^{-1}\}$ and $f(x) = \alpha x + \ln \alpha + 0.5 - \alpha$. We point out that if $x \geq \frac{\alpha - \ln \alpha - 0.5}{\alpha} + 0.1$, then $f(x) \geq 0.1\alpha$. The goal of this subsection is to build the following lemma.

**Lemma 4 (restated).** *If $\varkappa \geq \frac{\alpha - \ln \alpha - 0.5}{\alpha} + 0.1$, then conditioned on $\mathcal{F}_{\Lambda - f(\varkappa)}$,*

$$\mathcal{R}_B^{\mathrm{LSA}}(T) \leq \frac{9 \cdot \sqrt[8\alpha]{2}}{\sqrt[8\alpha]{2} - 1} \cdot \sum_{i \in B} \exp\left(-\frac{\lambda_i \Delta_i^2}{10} + \varkappa/4\right).$$

To prove Lemma 4, we make use of Lemma 16, Lemma 17 and Corollary 18, and defer their proofs in the later part of this subsection.

For any arm $i \in B$ and $\varkappa$, we define the event $\mathcal{M}_{i,\varkappa}$ by

$$\mathcal{M}_{i,\varkappa} \stackrel{\text{def}}{=} \left\{\forall t \in [1, \lambda_i], |\widehat{\Delta}_{i,t} - \Delta_i| \leq \sqrt{\frac{\lambda_i \Delta_i^2/5 - \varkappa/2 + \frac{1}{4\alpha} \ln \frac{\lambda_i}{t}}{t}}\right\}.$$

Intuitively, $\mathcal{M}_{i,\varkappa}$ requires that the estimation error of $\Delta_i$ during any time of the algorithm stays within a small band that is parameterized by the quality parameter $\varkappa$. The following Lemma 16 gives a lower bound on $\mathcal{M}_{i,\varkappa}$.

**Lemma 16.** *For any arm $i \in B$ and $\varkappa$, it holds that $\mathbb{P}(\mathcal{M}_{i,\varkappa}) \geq 1 - \frac{4 \cdot \sqrt[8\alpha]{2}}{\sqrt[8\alpha]{2} - 1} \exp\left(-\frac{\lambda_i \Delta_i^2}{10} + \varkappa/4\right).$*

The following Lemma 17 shows that $\mathcal{M}_{i,\varkappa}$ together with $\mathcal{F}_{\Lambda - f(\varkappa)}$ guarantees that arm $i$ is explored by enough queries.

**Lemma 17.** *For any arm $i \in B$ and $\varkappa \geq \frac{\alpha - \ln \alpha - 0.5}{\alpha}$, conditioning on $\mathcal{M}_{i,\varkappa} \wedge \mathcal{F}_{\Lambda - f(\varkappa)}$, we have that $T_i(T) \geq \lambda_i/20$.*

A corollary of Lemma 17 is as follows.

**Corollary 18.** *For any arm $i \in B$ and $\varkappa \geq \frac{\alpha - \ln \alpha - 0.5}{\alpha}$, we have*

$$\mathbb{P}(T_i(T) < \lambda_i/20 \mid \mathcal{F}_{\Lambda - f(\varkappa)}) \leq \frac{\mathbb{P}(\overline{\mathcal{M}}_{i,\varkappa})}{\mathbb{P}(\mathcal{F}_{\Lambda - f(\varkappa)})}.$$

We are now ready to give an upper bound for the contribution of the arms in $B$ to the aggregate regret of Algorithm 1.

*Proof.* Let $i$ be an arbitrary arm in $B$. Since $\mathbb{P}(\overline{\mathcal{E}}_i(T) \mid \mathcal{F}_{\Lambda - f(\varkappa)}) \leq 1$, it suffices to prove that

$$\mathbb{P}(\overline{\mathcal{E}}_i(T) \mid \mathcal{F}_{\Lambda - f(\varkappa)}) \leq \frac{9 \cdot \sqrt[8\alpha]{2}}{\sqrt[8\alpha]{2} - 1} \exp\left(-\frac{\lambda_i \Delta_i^2}{10} + \varkappa/4\right) \tag{22}$$

whenever $\frac{\sqrt[8\alpha]{2}}{\sqrt[8\alpha]{2}-1} \exp\left(-\frac{\lambda_i \Delta_i^2}{10} + \varkappa/4\right) \leq 1/9$. Then the lemma follows by summing up the inequality for all arms in $B$.

Notice that

$$\mathbb{P}(\overline{\mathcal{E}}_i(T) \mid \mathcal{F}_{\Lambda - f(\varkappa)})$$
$$= \mathbb{P}(\overline{\mathcal{E}}_i(T) \mid T_i \geq \lambda_i/20, \mathcal{F}_{\Lambda - f(\varkappa)}) \mathbb{P}(T_i \geq \lambda_i/20 \mid \mathcal{F}_{\Lambda - f(\varkappa)})$$
$$+ \mathbb{P}(\overline{\mathcal{E}}_i(T) \mid T_i < \lambda_i/20, \mathcal{F}_{\Lambda - f(\varkappa)}) \mathbb{P}(T_i < \lambda_i/20 \mid \mathcal{F}_{\Lambda - f(\varkappa)})$$
$$\leq \mathbb{P}(\overline{\mathcal{E}}_i(T) \mid T_i \geq \lambda_i/20, \mathcal{F}_{\Lambda - f(\varkappa)}) + \mathbb{P}(T_i < \lambda_i/20 \mid \mathcal{F}_{\Lambda - f(\varkappa)}). \quad (23)$$

We first focus on the first term of (23), and note that

$$\mathbb{P}(\overline{\mathcal{E}}_i(T) \mid T_i \geq \lambda_i/20, \mathcal{F}_{\Lambda - f(\varkappa)})$$
$$= \frac{\mathbb{P}(\overline{\mathcal{E}}_i(T) \wedge \mathcal{F}_{\Lambda - f(\varkappa)} \mid T_i \geq \lambda_i/20)}{\mathbb{P}(\mathcal{F}_{\Lambda - f(\varkappa)} \mid T_i \geq \lambda_i/20)}$$
$$\leq \frac{\mathbb{P}(\overline{\mathcal{E}}_i(T) \mid T_i \geq \lambda_i/20)}{\mathbb{P}(\mathcal{F}_{\Lambda - f(\varkappa)} \mid T_i \geq \lambda_i/20)} = \frac{\mathbb{P}(\overline{\mathcal{E}}_i(T) \wedge T_i \geq \lambda_i/20)}{\mathbb{P}(\mathcal{F}_{\Lambda - f(\varkappa)} \wedge T_i \geq \lambda_i/20)}$$
$$= \frac{\mathbb{P}(\overline{\mathcal{E}}_i(T) \wedge T_i \geq \lambda_i/20)}{(1 - \mathbb{P}(T_i < \lambda_i/20 \mid \mathcal{F}_{\Lambda - f(\varkappa)})) \cdot \mathbb{P}(\mathcal{F}_{\Lambda - f(\varkappa)})}. \quad (24)$$

Then plugging (24) into (23), we derive

$$\mathbb{P}(\overline{\mathcal{E}}_i(T) \mid \mathcal{F}_{\Lambda - f(\varkappa)}) \leq \frac{\mathbb{P}(\overline{\mathcal{E}}_i(T) \wedge T_i \geq \lambda_i/20)}{(1 - \mathbb{P}(T_i < \lambda_i/20 \mid \mathcal{F}_{\Lambda - f(\varkappa)})) \cdot \mathbb{P}(\mathcal{F}_{\Lambda - f(\varkappa)})} + \mathbb{P}(T_i < \lambda_i/20 \mid \mathcal{F}_{\Lambda - f(\varkappa)}). \quad (25)$$

Using Chernoff-Hoeffding Inequality (Proposition 5), we have

$$\mathbb{P}(\overline{\mathcal{E}}_i(T) \wedge T_i \geq \lambda_i/20) = \sum_{t=\lceil \lambda_i/20 \rceil}^{+\infty} \mathbb{P}(\overline{\mathcal{E}}_i(T) \mid T_i = t) \mathbb{P}(T_i = t)$$
$$\leq \sum_{t=\lceil \lambda_i/20 \rceil}^{+\infty} \mathbb{P}(T_i = t) \cdot 2 \exp(-\lambda_i \Delta_i^2/10) \leq 2 \exp(-\lambda_i \Delta_i^2/10). \quad (26)$$

Moreover, by Lemma 3 and the fact that $f(\varkappa) \geq 0.1\alpha$ for $\varkappa \geq \frac{\alpha - \ln \alpha - 0.5}{\alpha} + 0.1$, we have

$$\mathbb{P}(\mathcal{F}_{\Lambda - f(\varkappa)}) \geq 1 - \exp(-40 f(k)/\alpha) \geq 1 - \exp(-4) \geq 0.9. \quad (27)$$

Combining (27) with Corollary 18 and Lemma 16, we have

$$\mathbb{P}(T_i < \lambda_i/20 \mid \mathcal{F}_{\Lambda - f(\varkappa)})$$
$$\leq \frac{\mathbb{P}(\overline{\mathcal{M}}_{i,\varkappa})}{\mathbb{P}(\mathcal{F}_{\Lambda - f(\varkappa)})} \leq \frac{\frac{4 \cdot \sqrt[8\alpha]{2}}{\sqrt[8\alpha]{2} - 1} \exp\left(-\frac{\lambda_i \Delta_i^2}{10} + \varkappa/4\right)}{0.9}$$
$$\leq \frac{4.5 \cdot \sqrt[8\alpha]{2}}{\sqrt[8\alpha]{2} - 1} \exp\left(-\frac{\lambda_i \Delta_i^2}{10} + \varkappa/4\right). \quad (28)$$

Putting together (25), (26), (27), and (28), we obtain

$$\mathbb{P}(\overline{\mathcal{E}}_i(T) \mid \mathcal{F}_{\Lambda - f(\varkappa)})$$
$$\leq \frac{2 \exp(-\lambda_i \Delta_i^2/10)}{\left(1 - \frac{4.5 \cdot \sqrt[8\alpha]{2}}{\sqrt[8\alpha]{2} - 1} \exp\left(-\frac{\lambda_i \Delta_i^2}{10} + \varkappa/4\right)\right) \cdot 0.9} + \frac{4.5 \cdot \sqrt[8\alpha]{2}}{\sqrt[8\alpha]{2} - 1} \exp\left(-\frac{\lambda_i \Delta_i^2}{10} + \varkappa/4\right)$$
$$\leq \left(\frac{2}{(1 - 4.5/9) \cdot 0.9} + \frac{4.5 \cdot \sqrt[8\alpha]{2}}{\sqrt[8\alpha]{2} - 1}\right) \exp\left(-\frac{\lambda_i \Delta_i^2}{10} + \varkappa/4\right)$$
$$\leq \frac{9 \cdot \sqrt[8\alpha]{2}}{\sqrt[8\alpha]{2} - 1} \exp\left(-\frac{\lambda_i \Delta_i^2}{10} + \varkappa/4\right),$$

where the second inequality follows from our assumption that $\frac{\sqrt[8\alpha]{2}}{\sqrt[8\alpha]{2} - 1} \exp\left(-\frac{\lambda_i \Delta_i^2}{10} + \varkappa/4\right) \leq 1/9$. This proves (22) and therefore the lemma. $\qquad \square$

### D.3.1 Proof of Lemma 16

**Lemma 16 (restated).** *For any arm $i \in B$ and $\varkappa$, it holds that $\mathbb{P}(\mathcal{M}_{i,\varkappa}) \geq 1 - \frac{4 \cdot \sqrt[8]{2}}{\sqrt[8]{2}-1} \exp\left(-\frac{\lambda_i \Delta_i^2}{10} + \varkappa/4\right)$.*

In order to estimate the probability of $\mathcal{M}_{i,\varkappa}$, we introduce a more general lemma as follows and Lemma 16 becomes a simple corollary of Lemma 19.

**Lemma 19.** *(Variable Confidence Level Bound) Let $X_1, \ldots, X_L$ be i.i.d. random variables supported on $[0,1]$ with mean $\mu$. For any $a > 0$ and $b > 0$, it holds that*

$$\mathbb{P}\left(\forall t \in [1, L], \left|\frac{1}{t}\sum_{i=1}^{t} X_i - \mu\right| \leq \sqrt{\frac{a + b\ln(L/t)}{t}}\right) \geq 1 - \frac{2^{b/2+2}}{2^{b/2} - 1} \exp(-a/2).$$

Now we only need to prove Lemma 19.

*Proof of Lemma 19.* Let $l = \lfloor \log_2 L \rfloor$. By Chernoff-Hoeffding Inequality (Proposition 5), we have for any $t \in \{1, 2, 4, \ldots, 2^l\}$,

$$\mathbb{P}\left(\left|\frac{1}{t}\sum_{i=1}^{t} X_i - \mu\right| \leq \frac{1}{2}\sqrt{\frac{a + b\ln\frac{L}{t}}{t}}\right) \geq 1 - 2\exp(-a/2) \cdot \frac{t^{b/2}}{L^{b/2}}.$$

Via a union bound and using the fact that $2^{l+1} \leq 2L$, we get

$$\mathbb{P}\left(\forall t \in \{1, 2, 4, \ldots, 2^l\}, \left|\frac{1}{t}\sum_{i=1}^{t} X_i - \mu\right| \leq \frac{1}{2}\sqrt{\frac{a + b\ln\frac{L}{t}}{t}}\right)$$

$$\geq 1 - 2\exp(-a/2) \cdot \sum_{i=0}^{l} \frac{2^{bi/2}}{L^{b/2}} \geq 1 - \frac{2^{b/2+1}}{2^{b/2} - 1}\exp(-a/2). \quad (29)$$

By Hoeffding's Maximal Inequality (Proposition 7), we have for any $t \in \{1, 2, 4, \ldots, 2^l\}$,

$$\mathbb{P}\left(\forall j \in [1, \min\{t, L-t\}], |X_{i,t+1} + \cdots + X_{i,t+j} - j\mu| \leq \frac{1}{2}\sqrt{t\left(a + b\ln\frac{L}{t}\right)}\right)$$

$$\geq 1 - 2\exp(-a/2) \cdot \frac{t^{b/2}}{L^{b/2}}.$$

Again via a union bound and using the fact that $2^{l+1} \leq 2L$, we get

$$\mathbb{P}\left(\forall t \in \{1, 2, 4, \ldots, 2^l\}, \forall j \in [1, \min\{t, L-t\}], |X_{i,t+1} + \cdots + X_{i,t+j} - j\mu| \leq \frac{1}{2}\sqrt{t\left(a + b\ln\frac{L}{t}\right)}\right)$$

$$\geq 1 - \frac{2^{b/2+1}}{2^{b/2} - 1}\exp(-a/2). \quad (30)$$

Combining (29) and (30), and using a union bound, we have with probability at least $1 - \frac{2^{b/2+2}}{2^{b/2}-1}\exp(-a/2)$ uniformly over all $t \in \{1, 2, 4, \ldots, 2^l\}$ and $j \in [1, \min\{t, \lambda_i - t\}]$ that

$$|X_1 + \cdots + X_{t+j} - (t+j)\mu| \leq \sqrt{t\left(a + b\ln\frac{L}{t}\right)}.$$

Dividing both sides of the above inequality by $(t+j)$, we complete the proof of this lemma.

$\square$

### D.3.2 Proof of Lemma 17 and Corollary 18

**Lemma 17 (restated).** *For any arm $i \in B$ and $\varkappa \geq \frac{\alpha - \ln \alpha - 0.5}{\alpha}$, conditioning on $\mathcal{M}_{i,\varkappa} \wedge \mathcal{F}_{\Lambda - f(\varkappa)}$, we have that $T_i(T) \geq \lambda_i/20$.*

*Proof.* Fix an arm $i \in B$ and $\varkappa \geq \frac{\alpha - \ln \alpha - 0.5}{\alpha}$. We now condition on $\mathcal{M}_{i,\varkappa} \wedge \mathcal{F}_{\Lambda - f(\varkappa)}$ and prove this lemma by contradiction. Suppose for contradiction that we have $t < \lambda_i/20$. Notice that

$$
\begin{aligned}
\xi_{i,t} &\leq \alpha t \left( \Delta_i + \sqrt{\frac{\lambda_i \Delta_i^2/5 - \varkappa/2 + \frac{1}{4\alpha} \ln \frac{\lambda_i}{t}}{t}} \right)^2 + \ln \sqrt{t} \\
&\leq \alpha \left( \sqrt{\frac{\lambda_i}{20}} \Delta_i + \sqrt{\lambda_i \Delta_i^2/5 - \varkappa/2 + \frac{1}{4\alpha} \ln \frac{\lambda_i}{t}} \right)^2 + \ln \sqrt{t} \\
&\leq \alpha \left( 0.5 \lambda_i \Delta_i^2 - \varkappa + \frac{1}{2\alpha} \ln \frac{\lambda_i}{t} \right) + \ln \sqrt{t} \\
&\leq \alpha (0.5 \lambda_i \Delta_i^2 - \varkappa) + \ln \sqrt{\lambda_i}, \tag{31}
\end{aligned}
$$

It is easy to verify that $x = (\alpha \lambda_i)^{-\frac{1}{2}}$ is the minimum of the function $0.5 \alpha \lambda_i x^2 + \ln x^{-1}$ when $x > 0$. Hence, we have

$$
\ln \sqrt{\lambda_i} + \ln \sqrt{\alpha} + 0.5 = 0.5 \alpha \lambda_i (\alpha \lambda_i)^{-1} + \ln (\alpha \lambda_i)^{\frac{1}{2}} \leq 0.5 \alpha \lambda_i \Delta_i^2 + \ln \Delta_i^{-1}.
$$

Therefore,

$$
(31) \leq \alpha \lambda_i \Delta_i^2 + \ln \Delta_i^{-1} - \alpha - (\alpha \varkappa + \ln \alpha + 0.5 - \alpha).
$$

Finally, since $\Lambda = \alpha \lambda_i \Delta_i^2 + \ln \Delta_i^{-1} - \alpha$ for all $i \in B$ by definition, the last inequality yields $\xi_{i,t} \leq \Lambda - f(\varkappa)$, which contradicts the assumption that $\mathcal{F}_{\Lambda - f(\varkappa)}$ is true. $\square$

**Corollary 18 (restated).** *For any arm $i \in B$ and $\varkappa \geq \frac{\alpha - \ln \alpha - 0.5}{\alpha}$, we have*

$$
\mathbb{P}(T_i(T) < \lambda_i/20 \mid \mathcal{F}_{\Lambda - f(\varkappa)}) \leq \frac{\mathbb{P}(\overline{\mathcal{M}}_{i,\varkappa})}{\mathbb{P}(\mathcal{F}_{\Lambda - f(\varkappa)})}.
$$

*Proof.* Note that

$$
\mathbb{P}(T_i(T) < \lambda_i/20 \mid \mathcal{F}_{\Lambda - f(\varkappa)}) = 1 - \frac{\mathbb{P}(T_i(T) \geq \lambda_i/20 \wedge \mathcal{F}_{\Lambda - f(\varkappa)})}{\mathbb{P}(\mathcal{F}_{\Lambda - f(\varkappa)})}.
$$

Further by Lemma 17, we obtain

$$
\mathbb{P}(T_i(T) < \lambda_i/20 \mid \mathcal{F}_{\Lambda - f(\varkappa)}) \leq 1 - \frac{\mathbb{P}(\mathcal{M}_{i,\varkappa} \wedge \mathcal{F}_{\Lambda - f(\varkappa)})}{\mathbb{P}(\mathcal{F}_{\Lambda - f(\varkappa)})} = \frac{\mathbb{P}(\overline{\mathcal{M}}_{i,\varkappa} \wedge \mathcal{F}_{\Lambda - f(\varkappa)})}{\mathbb{P}(\mathcal{F}_{\Lambda - f(\varkappa)})} \leq \frac{\mathbb{P}(\overline{\mathcal{M}}_{i,\varkappa})}{\mathbb{P}(\mathcal{F}_{\Lambda - f(\varkappa)})},
$$

which concludes the proof of this corollary. $\square$

## E  The Lower Bound

In this section we discuss the regret lower bound of *any* algorithm. For any sequence of $K$ gaps $\Delta_1, \ldots, \Delta_K > 0$, let $\mathcal{I}_{\Delta_1, \ldots, \Delta_K}$ denote the set of instances of the problem where the gap between $\theta_i$ and $\theta$ is $\Delta_i$ for every arm $i \in [K]$. We now show a parameter dependent lower bound of the aggregate regret when the time horizon $T \geq K$.

**Theorem 20.** *Let $(\Delta_1, \ldots, \Delta_K) \in (0, 1/4]^K$ be a sequence of gaps. Then for any algorithm $\mathbb{A}$ and time horizon $T \geq K$, there exists an instance $I \in \mathcal{I}_{\Delta_1, \ldots, \Delta_K}$ such that*

$$
\mathcal{R}^{\mathbb{A}}(I; T) \geq \frac{1}{4} \mathscr{P}_{16}(\{\Delta_i\}_{i \in S}, T).
$$

*Proof.* Let $\mathcal{B}(\mu)$ denote the Bernoulli distribution with mean $\mu$. We use $\Delta_i(I)$ to denote the gap between arm $i$ and the threshold $\theta$, given the instance $I$. For any algorithm $\mathbb{A}$ and instance $I$, let $\mathcal{D}_{I\mathbb{A}}$ denote the probability space induced by $I$ and $\mathbb{A}$. We use $\mathbb{P}_{I\mathbb{A}}$ to denote the measure of the probability space $\mathcal{D}_{I\mathbb{A}}$, and use $\mathbb{E}_{I\mathbb{A}}[\cdot]$ to denote the expectation with respect to $\mathbb{P}_{I\mathbb{A}}$. When clear from the context, the reference to $\mathbb{A}$ is omitted.

We fix the threshold $\theta = 1/2$. To prove the theorem, it suffices to prove that there exist $2^K$ instances $I_0, \ldots, I_{2^K-1} \in \mathcal{I}_{\Delta_1, \ldots, \Delta_K}$ such that

$$\max_{0 \le j < 2^K} \mathcal{R}^{\mathbb{A}}(I_j; T) \ge \frac{1}{4} \mathscr{P}_{16}(\{\Delta_i\}_{i \in S}, T).$$

We now define these instances explicitly. Suppose the binary representation of $j$ is denoted by $\overline{a_1^j \cdots a_K^j}$. Then for any arm $i$ in $I_j$, the associated distribution is $\mathcal{B}(1/2 + a_i^j \Delta_i)$. Thus the distribution associated with $I_j$ can be represented by the product distribution

$$\mathcal{B}(1/2 + a_1^j \Delta_1) \otimes \cdots \otimes \mathcal{B}(1/2 + a_K^j \Delta_K).$$

First, we note that

$$
\begin{aligned}
\max_{0 \le j < 2^K} \mathcal{R}^{\mathbb{A}}(I_j; T) &= \max_{0 \le j < 2^K} \sum_{i=1}^{K} \mathbb{P}_{I_j}(\overline{\mathcal{E}}_i(T)) \\
&\ge \frac{1}{2^K} \sum_{j=0}^{2^K-1} \sum_{i=1}^{K} \mathbb{P}_{I_j}(\overline{\mathcal{E}}_i(T)).
\end{aligned}
\tag{32}
$$

By counting $\mathbb{P}_{I_j}(\overline{\mathcal{E}}_i(T))$ twice and then reordering, we have

$$(32) = \frac{1}{2^{K+1}} \sum_{i=1}^{K} \sum_{j=0}^{2^K-1} \left( \mathbb{P}_{I_j}(\overline{\mathcal{E}}_i(T)) + \mathbb{P}_{I_{j \oplus 2^{i-1}}}(\overline{\mathcal{E}}_i(T)) \right), \tag{33}$$

where $\oplus$ denotes the binary XOR operation. Now from Proposition 8, we get that for $i \in [K]$,

$$
\begin{aligned}
&\mathbb{P}_{I_j}(\overline{\mathcal{E}}_i(T)) + \mathbb{P}_{I_{j \oplus 2^{i-1}}}(\overline{\mathcal{E}}_i(T)) \\
&\ge \frac{1}{2} \exp(-D_{\mathrm{KL}}(\mathbb{P}_{I_j} \parallel \mathbb{P}_{I_{j \oplus 2^{i-1}}})) \\
&\overset{(a)}{=} \frac{1}{2} \exp(- \mathbb{E}_{I_j}[x_i] \cdot D_{\mathrm{KL}}(\mathcal{B}(1/2 + a_i^j \Delta_i) \parallel \mathcal{B}(1/2 + a_i^{j \oplus 2^{i-1}} \Delta_i))) \\
&= \frac{1}{2} \exp\left( - \mathbb{E}_{I_j}[x_i] \cdot 2\Delta_i \ln\left( 1 + \frac{2\Delta_i}{1/2 - \Delta_i} \right) \right) \\
&\overset{(b)}{\ge} \frac{1}{2} \exp\left( -\frac{4\,\mathbb{E}_{I_j}[x_i]\Delta_i^2}{1/2 - \Delta_i} \right) \\
&\ge \frac{1}{2} \exp(-16\,\mathbb{E}_{I_j}[x_i]\Delta_i^2),
\end{aligned}
\tag{34}
$$

where (a) follows from standard divergence decomposition (Proposition 9) and (b) follows from $\Delta_i \le 1/4$. Finally, plugging (34) into (33), we have

$$
\begin{aligned}
&\max_{0 \le j < 2^K} \mathcal{R}^{\mathbb{A}}(I_j; T) \\
&\ge \frac{1}{2^{K+1}} \sum_{j=0}^{2^K-1} \sum_{i=1}^{K} \frac{1}{2} \exp(-16\,\mathbb{E}_{I_j}[x_i]\Delta_i^2) \\
&\ge \frac{1}{2^{K+1}} \sum_{j=0}^{2^K-1} \min_{\substack{x_1+\cdots+x_K=T \\ x_1,\ldots,x_K \ge 0}} \sum_{i=1}^{K} \frac{1}{2} \exp(-16 x_i \Delta_i^2) \\
&= \min_{\substack{x_1+\cdots+x_K=T \\ x_1,\ldots,x_K \ge 0}} \sum_{i=1}^{K} \frac{1}{4} \exp(-16 x_i \Delta_i^2)
\end{aligned}
$$

$$= \frac{1}{4} \mathscr{P}_{16}(\{\Delta_i\}_{i \in S}, T),$$

which concludes the proof of this theorem. □

# F  Additional Experimental Results

## F.1  Comparison with APT$(0)$

The following figure demonstrates the experimental results when LSA is compared with APT$(0)$ and some other algorithms under the scenario that there is no indifference zone and simple regret is used. Please refer to Section 5 for different setting details.

Figure 2: Simple regret on a logarithmic scale for different settings.

## F.2  T-tests

In order to statistically compare our algorithm and other algorithms, we perform two-tailed paired t-tests between our algorithm and other algorithms respectively on 5000 independent runs. When performing t-tests, we set $T = 1000$, $T = 40000$, and $T = 100000$ in Setup 1, 2 and 3 respectively. The null hypothesis is that our algorithm and other algorithms have the same mean. The p-values are listed in the following table.

| Setup | P-values | | | | |
|---|---|---|---|---|---|
| | APT(0) | APT(.025) | APT(.05) | APT(.1) | UCBE(-1) |
| Setup 1 | 1.4e-32 | 0.60 | 0.95 | 1.3e-5 | 0 |
| Setup 2 | 0 | 8.9e-21 | 1.5e-78 | 4.0e-160 | 0 |
| Setup 3 | 0 | 7.8e-28 | 3.6e-88 | 1.5e-192 | 0 |

| Setup | P-values | | | | |
|---|---|---|---|---|---|
| | UCBE(0) | UCBE(4) | Opt-KG(1,1) | Opt-KG(.5,.5) | Uniform |
| Setup 1 | 2.5e-69 | 4.1e-225 | 7.8e-3 | 5.6e-8 | 1.1e-295 |
| Setup 2 | 0 | 1.6e-251 | 2.9e-26 | 4.6e-46 | 0 |
| Setup 3 | 0 | 2.0e-157 | 2.1e-24 | 6.5e-36 | 0 |

Table 1: T-test results between our algorithm and other algorithms in Setup 1, 2 and 3

## F.3  Experimental Results for More Setups

We present additional experimental results with other settings of $\{\theta_i\}_{i=1}^{K}$.

4. (geometric progression). $K = 10$; $\theta_{1:4} = 0.4 - 0.2^{(1:4)}, \theta_5 = 0.45, \theta_6 = 0.55$, and $\theta_{7:10} = 0.6 + 0.2^{5-(1:4)}$ (see Setup 4 in Figure 3).

5. (two-group setting II). $K = 100$; $\theta_{1:50} = 0.4$ and $\theta_{51:100} = 0.51$ (see Setup 5 in Figure 3).

6. (one-side group). $K = 10$; $\theta_{1:10} = 0.4 + (i-1)/100$ (see Setup 6 in Figure 3).

Figure 3: Average aggregate regret on a logarithmic scale for additional set of settings.

## F.4   Robustness of the Tuning Parameter in LSA

To test the robustness of our algorithm, we show the results of our algorithm for Setup 1 when varying $\alpha$ in Figure 4. We find that the performance is very consistent with different choices of $\alpha$. The differences are marginal and not statistically significant. For simplicity, in our experiments in the main text, we use $\alpha = 1.35$ as default (black curve in the figure).

Figure 4: Average aggregate regret on a logarithmic scale of LSA($\alpha$) on Setup 1 for different $\alpha$.