[Reviews · NeurIPS 2019]

Reviewer 1



Overview: This paper studies the thresholding bandit problem, under an alternative definition of regret to that which has been considered previously in the literature. The authors then provide an algorithm specifically tailored to minimizing this regret (up to some potentially enormous constant factors). They provide an analysis of the regret of this algorithm and demonstrate experimentally that it outperforms algorithms developed for minimizing a different regret definition. However, I would have liked to see more discussion and comparison of the performance of the different algorithms under different regret definitions and details of the parameter choice for their algorithm. Comments: - I don’t find the motivation for the need to consider aggregate regret particularly convincing. - Given that there is a lot of discussion about the simple regret, it would be good to include a definition of it, and discussion of how they relate to each other, e.g. does one imply the other? - In the discussion of Locatelli et al. (2016), it is stated that the algorithm is parameter free then that it takes parameter epsilon. - The uniform sampling method introduced in line 69 seems very naive so I am surprised it is only a factor of K worse than the proposed method. - Although space is tight, I would have liked to see at least some discussion of the best arm identification problem. - I think the discussion of the optimization problem in the case where the gaps are known is nice. It is also nice to see how this motivates the algorithm used. - The outline of the proof of Theorem 1 would be much better placed next to Theorem 1 rather than its current position in the introduction (lines 107-118). It also seems like a fairly standard analysis with the only interesting aspect the avoidance of a union bound. - A lot of the constants seem quite arbitrary (e.g. why do we need alpha<=8 and T>40?). - The constant in remark 2 is enormous. The authors should at least attempt to optimize it to give a meaningful result rater than making ‘no effort’ to do so. - How should alpha be chosen? In remark 2 it is 1/20, whereas in the experiments it is (somewhat arbitrarily) 1.35. - How are a and b chosen in Lemma 19? It seems somewhat un-intuitive that for large values of b, the distance between the empirical mean and expected value is large but the constant in front of the exponentially small probability more or less stays the same. - why is this analysis applicable to other areas but the analysis of MOSS not (line 267)? - I would like to see the aggregate regret of the APT algorithm of Locatelli et al(2016) and the simple regret of this algorithm for comparison. - In Figure 1, why is APT given with lots of different parameter settings where as their algorithm is only given with one. - In Figure 3 of Appendix F.3, the range of alphas considered is quite small. What happens if we set alpha=1/20 as in remark 2? - It would be good to see confidence bounds on the experimental results to see if the differences in performance are significant. - I would like to experimental results for the simple regret considered by Locatelli et al (2016) . Clarity: Often, the paper is difficult to read and care should be put into proof-reading to eliminate typos/grammatical errors. The introduction could also be significantly improved. After rebuttal: Thank you to the authors for providing a detailed rebuttal that addressed most of my concerns. I was particularly pleased to see a discussion of the relationships between the two regrets, the difference between their analysis and that of MOSS and the significance of the suboptimality of the uniform sampling method. I have therefore raised my score. For the final version, I hope that the authors will work on the readability of the paper and add more details about the choice of alpha.

Reviewer 2



Significance and originality : There is a huge number (and it's growing) of small variations of settings for pure exploration in MABs, and this paper adds to them by changing the notion of regret in thresholding bandits. The algorithm LSA is quite similar in form the APT of Locatelli et. al . That said, LSA is well motivated (with the optimisation problem), and the authors provide a well-rounded analysis of the problem with non-trivial methods, which is always interesting for experts. Quality and clarity : The paper is quite complete and thorough, answering spontaneously many natural questions (e.g. illustrating how the uniform sampling algorithm behaves, how to tune alpha, ...). Almost all mathematical operations are well-motivated, which is a pleasure for the reader, and the mathematical statements are followed by insightful comments. The experiments are quite complete too and seem to honestly show that their algorithm behaves well in practice. There is however a caveat: the theoretical dependence on \alpha is a little worrying, as the lambda_i's depend themselves on \alpha. Therefore, Remark 2 is not completely clear, and the claim of instance optimality is not properly proven. I do not think this is a bad problem, but it should be clarified. In general, I have a feeling that the exposition in Section 4 could be improved, as it confronts very suddenly the reader with some heavy notations that hinder readability. All in all, I think that this is a strong contribution, currently with an exposition problem in a critical part of the paper. Some typos : l32 : z_i = 1 iff \theta_i \geq 1/2 overall the description of the problem seems weird to me, why not just say we want to predict whether \theta_i \geq 1/2, without talking about the z_i ? l.90 : why is that ? l.162 : “Note that this is all AN algorithm can do […]” (the AN is missing). Why is this ? l.170 : approximates P_c^\star well l.173 : I think you mean max ? l.174 : function on x <- function of x l.182 : value <- values l.203 : alpha = 1/ 20 <= alpha = 1/10 ? l.206 : demonstrated <- proved, or stated ?? l268 : MOSS is not asymptotically optimal in the usual sense, although it is minimax optimax. l302 : the delta’s in the definition of H are not the delta's in the TBP l332 : the celebrated Hoeffding’s maximal inequality <- Hoeffding’s celebrated maximal inequality l.408 : applies <- apply to l454 : performing THE uniform Post rebuttal edit. The setting and the algorithm are interesting. However, the issue I find critical, about optimality, has not been answered properly in the author feedback and I maintain my score. To me, this is a potentially strong but incomplete work as long as the regret bounds are not more readable/comparable to lower bounds.

Reviewer 3



The paper considers the thresholding bandit problem, where aggregate regret is minimized instead of simple regret (errors on all arms, instead of at least one). A new algorithm is proposed for the slightly modified setting. The paper derives the aggregate regret bound for the algorithm, which roughly matches the lower bound provided in the appendix. Experiments on synthetic data show that the algorithm performs better than some existing algorithms (which were designed for slightly different variant of the problem). Minimizing aggregate regret (as opposed to simple regret) seems to make sense in some scenarios. Consequently, it does warrant additional analysis. Given that the existing algorithms were designed for different problems it is difficult to assess the improvement in performance (regret bound or empirical). The theoretical analysis contain sufficient new elements (including the variable confidence level bound) to be considered worthy of publication. The article is well written, and the split in content between the main submission and the appendix is also reasonable. Maybe, the lower bound could be stated in the main part as well. Update after the author response: I appreciate the authors’ effort, in particular the discussion on simple and aggregate regret and the additional experiments.

[Author Response · NeurIPS 2019]

We thank all the reviewers for their helpful and constructive comments. Here we respond to the major concerns. We
will fix the minor issues in the new version of the paper.

**On Remark 2 and the statement about optimality (R1 and R2).** We agree with R2 that $\lambda_i$'s depend on $\alpha$, and
Remark 2 only says that the LSA algorithm is instance-wise asymptotically optimal when $\alpha$ is chosen properly, e.g.
$1/20$. We did not prove the optimality for $\alpha > 1/20$, due to the factor $1/10$ in the upper bound of Theorem 1, which
arises from several places in the proof details, and is also responsible for the large constants in Remark 2. However,
we believe that this is only because of the limitation of our current proof techniques, and the algorithm might be
asymptotically optimal with much smaller constants for larger $\alpha$, which we leave as a future work. Empirical evaluation
suggests that the algorithm performs very well when $\alpha \in [1.15, 1.55]$. We will add this clarification and revise the
exposition of the statements following R2's suggestion.

**On comparison of LSA and APT (R1, R2, and R3).** $\mathrm{APT}(\epsilon = 0)$ may be intrigued by a hard arm (as it has to
correctly label all arms to achieve a small simple regret) and waste many samples, and cannot achieve the optimal
sample efficiency for the aggregate regret. On Line 60, we also argued that APT with $\epsilon > 0$ cannot achieve the optimal
aggregate regret. Indeed, in our experiments, $\mathrm{APT}(\epsilon = 0)$ performs consistently worse than LSA in all settings.

On the other hand, a simple corollary of our main theorem shows that, for the simple regret *with no* indifference zone
($\epsilon = 0$), our LSA achieves optimality up to a $\ln K$ factor in the budget $T$. Note that the optimality definition in (Locatelli
et al., 2016) also allows logarithmic slacks of $T$. In contrast, in our paper, we aim for the strict asymptotic optimality
up to only constant factors. We leave proving the strict asymptotic optimality and generalizing LSA to achieve the
optimal simple regret *with* indifference zone as an open direction for future research. We attach new experimental
results (Figure 1) to show that LSA performs better than $\mathrm{APT}(0)$ for the simple regret with no indifference zone.

**Other comments by R1.**

*Regarding the motivation of the problem.* In-
deed, the aggregate regret is more practically
relevant than the previously used simple
regret, as the aggregate regret corresponds to
the accuracy which practitioners care about,
while the simple regret is not empirically
meaningful. For example, we use the
binary labeling task in crowdsourcing as a

Figure 1: Simple regret on a logarithmic scale for different settings.

motivating example for the thresholding bandit problem with aggregate regret. The same problem (under the Bayesian
setting) and its application to crowdsourcing has been extensively studied in (Chen et al., 2015). The reviewer may
refer to (Chen et al., 2015) for more detailed information on applying our problem to crowdsourcing tasks.

*Regarding the definition of the simple regret and the relationship between the two regrets.* The simple regret is defined
On Line 25. By Markov's inequality, we know that an algorithm with $\delta$ aggregate regret has at most $\delta$ simple regret,
and an algorithm achieving $\delta$ simple regret has at most $K\delta$ aggregate regret.

*Regarding "parameter-free" in APT.* The APT algorithm is named by the authors of (Locatelli et al., 2016) to be
"Anytime Parameter-free Thresholding". In their setting, the $\epsilon$ indifference zone is given as a problem input and not
considered as a tuning parameter. However, $\epsilon$ has to be set by the user.

*Regarding the uniform sampling method,* being worse than the optimal algorithm by a factor of $\Omega(K)$ is very bad. It
means that the algorithm cannot distinguish the arms when making adaptive query decisions (which is indeed the case
for uniform sampling), and therefore is a trivial performance guarantee.

*Regarding the technical contribution,* we think our algorithm design and analysis technique to avoid the $\ln K$ caused
by union bounds is highly non-trivial. In addition, the variable confidence level technique is also an important technical
contribution. We believe that the overall technical contribution is significant, as also mentioned by both R2 and R3.

*Regarding the constants.* Since most of the contributions of this paper are theoretical, we do not focus on optimizing the
constants, which is usual in most theoretical work. Please refer to the first paragraph for the setting of $\alpha$ in Remark 2.

*Regarding Lemma 19,* we set $a = \lambda_i \Delta_i^2/5 - \varkappa/2$ and $b = 1/(4\alpha)$ to prove Lemma 16. We agree with the reviewer
that when $b \gg 1$, the confidence interval becomes longer while the error probability does not change much. In our
proof, we only use the lemma when $b = O(1)$. It is possible that the lemma could be further improved when $b \gg 1$.

*Regarding the analysis.* On Line 267, we note that our Lemma 19 is a generalization of Hoeffding's maximal inequality,
which is used in the analysis of MOSS. Our Lemma 19 can also replace the usage of Hoeffding's maximal inequality in
the analysis of MOSS, and may find other applications. However, the analysis of our algorithm is very different from
MOSS, despite them sharing the high-level intuition. We will make this clear in the next version of the paper.

*Regarding the statistical significance.* In Appendix F.1, we use T-tests to show that the confidence intervals of the
performances of LSA and other algorithms do not overlap in most settings.

[Meta-Review · NeurIPS 2019]

The reviews agree on the interest of the paper, and provide a good number of suggestions for improvements for the final version. In particular, the authors should explain better the motivation for their alternative definition of the regret, and (although most issues raised by the reviewers were addressed in the rebuttal, which tended to convince the reviewers) the criticism of one reviewer on optimality was not addressed properly answered to: improving this point would be a significant good point for the final version.